# STATISTICAL REJECTION SAMPLING IMPROVES PREFERENCE OPTIMIZATION

**Tianqi Liu**[*], **Yao Zhao**[†], **Rishabh Joshi**[†], **Misha Khalman**[†], **Mohammad Saleh**[†],
**Peter J. Liu**[†], **Jialu Liu**[*]
{tianqiliu,yaozhaoyz,rishabhjoshi,khalman,msaleh,
peterjliu,jialu}@google.com
Google Research[*], Google DeepMind[†]

## ABSTRACT

Improving the alignment of language models with human preferences remains an active research challenge. Previous approaches have primarily utilized online Reinforcement Learning from Human Feedback (RLHF). Recently, offline methods such as Sequence Likelihood Calibration (SLiC) and Direct Preference Optimization (DPO) have emerged as attractive alternatives, offering improvements in stability and scalability while maintaining competitive performance. SLiC refines its loss function using sequence pairs sampled from a supervised fine-tuned (SFT) policy, while DPO directly optimizes language models based on preference data, foregoing the need for a separate reward model. However, the maximum likelihood estimator (MLE) of the target optimal policy requires labeled preference pairs sampled from that policy. The absence of a reward model in DPO constrains its ability to sample preference pairs from the optimal policy. Meanwhile, SLiC can only sample preference pairs from the SFT policy. To address these limitations, we introduce a novel offline approach called *Statistical Rejection Sampling Optimization* (RSO) designed to source preference data from the estimated target optimal policy using rejection sampling, enabling a more accurate estimation of the optimal policy. We also propose a unified framework that enhances the loss functions used in both SLiC and DPO from a preference modeling standpoint. Through extensive experiments across diverse tasks, we demonstrate that RSO consistently outperforms both SLiC and DPO as evaluated by gold reward, Large Language Models (LLMs) and human raters.

## 1 INTRODUCTION

Recent advancements in Large Language Models (LLMs) (Brown et al., 2020; Touvron et al., 2023; Anil et al., 2023; OpenAI, 2023) have unlocked unprecedented capabilities in diverse tasks, such as programming and creative writing. Models are pre-trained on large unlabeled corpora and supervised fine-tuned (SFT) on various tasks (Wei et al., 2021; Chung et al., 2022). Subsequently, RLHF (Stiennon et al., 2020) enhances the alignment of large language models with human preferences. RLHF introduces notable complexities into the training process, including a reward model, a policy model, a reference policy, and a value model. It limits the maximum feasible size of a model due to memory constraints. Additionally, it is not stable during training. Recognizing these challenges, recent research has pioneered alternatives to RLHF. Notable among these are RRHF (Yuan et al., 2023), SLiC (Zhao et al., 2022; 2023) and DPO (Rafailov et al., 2023). These methodologies aim to more effectively align LLMs with human preferences while avoiding the complexities of reinforcement learning. Given supervised finetuning data $\mathcal{D}_{\text{sft}} = \{(x, y_{\text{ref}})\}$ and preference data $\mathcal{D}_{\text{hf}} = \{(x, y_w, y_l)\}$ where output text $y_w$ is preferred over $y_l$ on the same input text $x$, they directly fit the policy model on preference data in various ways. RRHF uses a trained reward model or human raters to compute rewards for multiple sequences generated from difference sources on the same prompt $x$, and then apply a ranking loss plus supervised fine-tuning loss. SLiC uses a contrastive ranking calibration loss plus a regularization loss

$$\mathcal{L}(\theta) = \max\left(0, \delta - \log \pi_\theta(y_w|x) + \log \pi_\theta(y_l|x)\right) - \lambda \log \pi_\theta(y_{\text{ref}}|x), \tag{1}$$

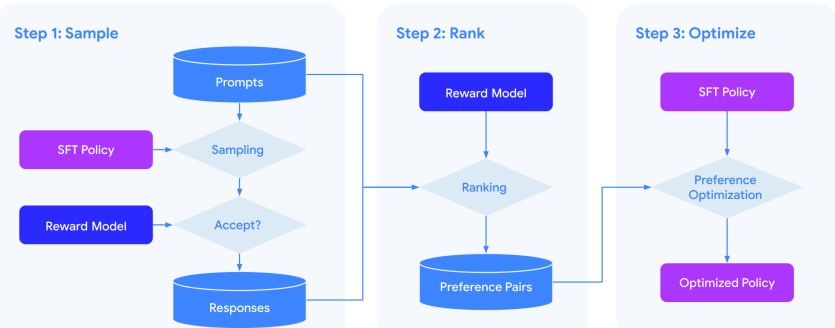

Figure 1: RSO first fits a pairwise reward-ranking model from human preference data. This model is later applied to generate preference pairs with candidates sampled from the optimal policy, followed by a preference optimization step to align sequence likelihood towards preferences.

where $\delta$ is a positive margin and $\pi_\theta$ is the learnable conditional probability function by a language model. SLiC either fits directly on human preference data or on preference data sampled from the SFT policy. DPO analyzes RLHF's objective function in the form of KL-regularized reward maximization, and analytically solves the optimal policy induced by a reward function. Based on the Bradley-Terry (BT) model (Bradley & Terry, 1952), DPO proposes an MLE to fit on human preference data directly and expresses the human preference probability in terms of only the optimal policy $\pi^*$ and reference policy $\pi_{\text{sft}}$:

$$p^*(y_1 \succ y_2|x) = \frac{1}{1 + \exp\left(\beta \log \frac{\pi^*(y_2|x)}{\pi_{\text{sft}}(y_2|x)} - \beta \log \frac{\pi^*(y_1|x)}{\pi_{\text{sft}}(y_1|x)}\right)} \tag{2}$$

where $\pi^*$ is the function to be estimated and $\beta$ is a hyparameter in RLHF objective.

Empirically, one can leverage observed preference pairs to approximate $p^*(y_1 \succ y_2|x)$. To estimate $\pi^*$ as a density estimation problem, the optimal way is to fit a policy model on collected preference pairs sampled from $\pi^*$. However, DPO uses the collected human preference data from other policies directly in all the experiments and lacks a study on the effect of sampling. Although they propose to sample pairs from the SFT policy and get them labeled by human, it is still not strictly MLE for the preference model due to the mismatch between the sampling distribution and $\pi^*$. In reality, it is very challenging to obtain human preference pairs directly sampled from $\pi^*$.

In this work, we address the above issues by constructing preference pairs from the approximated $\pi^*$ (Figure 1). Starting with a human preference dataset $\mathcal{D}_{\text{hf}}$ collected from other policies, we first train a pairwise reward-ranking model, then apply a statistical rejection sampling algorithm to generate response pairs sampled from optimal policy by using SFT policy and the pairwise reward-ranking model. After that, we label the sampled response pairs by the reward model. Then we fit the model on labeled pairs via classification loss. DPO claims that the language model is secretly a reward model, we show that the language model learns better from an explicit reward model because comparing between two responses (reward) is easier to learn than generating high quality responses (policy). Our statistical rejection sampling refers to the one in the statistical field (Neal, 2003). In RLHF works (Bai et al., 2022; Stiennon et al., 2020; Touvron et al., 2023), they usually refer to rejection sampling as best-of-N or top-k-over-N algorithm, where they sample a batch of N completions from a language model policy and then evaluate them across a reward model, returning the best one or the top k. This algorithm has the issue of reward hacking because it trusts the reward model too much without any regularization. In this paper we show that top-k-over-N is a special case of our statistical rejection sampling and it is critical to balance between the reward exploitation and regularization towards the SFT policy. To summarize, our contributions of this work are three-fold.

- we propose a scalable and easy-to-implement framework to learn from human preference data. We provide a comprehensive recipe among different choices of loss functions and preference pairs generation. We show the importance of the reward model instead of directly optimizing the model on the preference data.

- we unify DPO and SLiC statistically by showing that they vary by loss functions to fit on human preference data: DPO is a logistic regression on human preference data and SLiC is *almost* equivalent to a support vector machine (SVM) with hinge loss. We improve SLiC as the SVM counter part of DPO.
- we design a statistical rejection sampling algorithm to sample pairs from the estimated optimal policy and get them labeled by a pairwise reward-ranking model. The proposed sampling strategy is shown to be effective on several generative tasks.

## 2 PRELIMINARIES

**Learning from Human Feedback**  Several works (Ziegler et al., 2019; Zhao et al., 2023; Rafailov et al., 2023) show the significant improvement of conditional language generation by learning from human feedback data. All algorithms take two inputs:

- $\pi_{\text{sft}}(y|x)$: a supervised fine-tuned policy (SFT), where $x$ is the prompt and $y$ is the response.
- $\mathcal{D}_{\text{hf}} = \{x^{(i)}, y_w^{(i)}, y_l^{(i)}\}_{i=1}^N$: a human preference dataset that distinguishes the better response from the worse given the same prompt.

**KL-Constrained Reward Maximization Objective**  Starting with a reward function $r(x,y)$ and input prompt distribution $\mathcal{P}$, the DPO and RLHF optimizes for the following objective:

$$\max_\pi \mathbb{E}_{x\sim\mathcal{P}, y\sim\pi} \left[ r(x,y) \right] - \beta \mathbb{D}_{KL} \left[ \pi(y|x) || \pi_{\text{sft}(y|x)} \right] \tag{3}$$

**Optimal Policy**  DPO solves the optimal policy $\pi_r(y|x)$ that maximizes the above objective:

$$\pi_r(y|x) = \frac{1}{Z(x)} \pi_{\text{sft}}(y|x) \exp\left( \frac{1}{\beta} r(x,y) \right) \tag{4}$$

for all $x \in \mathcal{P}$, where $Z(x) = \sum_y \pi_{\text{sft}}(y|x) \exp\left( \frac{1}{\beta} r(x,y) \right)$ is the partition function. $\beta$ controls the balance between exploitation and exploration. When $\beta \to 0$, all probability mass will concentrate on the max reward with full exploitation. When $\beta \to \infty$, optimal policy will be the same as $\pi_{\text{sft}}$ with full exploration. Rearrange the Equation (4) we get

$$r(x,y) = \beta \log \frac{\pi_r(y|x)}{\pi_{\text{sft}}(y|x)} + \beta \log Z(x). \tag{5}$$

The Equation (4) and (5) establish the relation between optimal policy and the reward function. In reality, the final goal is to have a good policy for response generation and $\pi_r(y|x)$ is usually of more interest. The key is to effectively estimate the $\pi_r(y|x)$ from the human preference data.

**Preference Model**  Let the ground-truth reward function be $r^*$, then the optimal policy $\pi^*$ associated with $r^*$ can be represented by Equation (4). For two responses $(y_1, y_2)$ from the same input $x$, Bradley-Terry (BT) model (Bradley & Terry, 1952) assumes that

$$\mathbb{P}(y_1 \succ y_2 | x) = \sigma(r^*(x, y_1) - r^*(x, y_2)), \tag{6}$$

where $\mathbb{P}(y_1 \succ y_2 | x)$ represents the probability that response $y_1$ is preferred over $y_2$ give prompt $x$. Reusing Equation (5), we obtain Equation (2). If we leverage the human preference data to represent $\mathbb{P}(y_1 \succ y_2 | x)$, the estimation of $\pi^*$ can be viewed as a density estimation problem from the preference data. We will discuss different ways of estimating $\pi^*$ in Section 3.1.

**Reward Model**  We train a pairwise T5-XXL (Raffel et al., 2020) text-to-text reward-ranking model[1] $\rho_\psi(x, y_1, y_2)$ on $\mathcal{D}_{\text{hf}}$ to approximate $\mathbb{P}(y_1 \succ y_2 | x)$. $\rho_\psi(x, y_1, y_2)$ takes the text input as:

- "[CONTEXT] $\{x\}$ [SUMMARY A] $\{y_1\}$ [SUMMARY B] $\{y_2\}$" for summarization task

---

[1]SLiC demonstrates that pairwise reward model is preferred in RL-free learning. Our pairwise reward-ranking model has accuracy of 73.23% on the validation set of summarization task and 69.75% on the validation set of AI assistant task.

- "[CONTEXT] $\{x\}$ [RESPONSE A] $\{y_1\}$ [RESPONSE B] $\{y_2\}$" for AI assistant task

$\rho_\psi(x, y_1, y_2)$ outputs "A" or "B" as preferred one. We use the probability of decoding "A" as estimation of the preference probability $\hat{\mathbb{P}}(y_1 \succ y_2|x)$[2]. Suppose we have a baseline sequence $y_b$ with reward score 0, we can induce the reward score of any sequence $y$ as

$$r_\psi(x, y) = \text{logit}(\rho_\psi(x, y, y_b)), \tag{7}$$

where $\text{logit}(x) = \log(\frac{x}{1-x})$. This is a result of setting $y_1 = y$, $y_2 = y_b$, and $r^*(x, y_2) = 0$ in Equation (6), where we replace the win rate with the estimated one $\rho_\psi(x, y, y_b)$. Thus, "pointwise" reward score can be derived from a "pairwise" reward-ranking model with a baseline sequence[3].

## 3 RSO APPROACH

### 3.1 STATISTICAL ESTIMATION OF THE OPTIMAL POLICY $\pi^*$

Our proposed approach (Figure 1) takes inputs of SFT policy, reward-ranking model, and prompts. First we sample responses from the optimal policy through rejection sampling approach, then we fit a classification model on labeled preference pairs. To study the effectiveness of our approach, we consider a few options on loss and preference dataset construction. Given a preference dataset $\mathcal{D}_p = \{(x^{(i)}, y_w^{(i)}, y_l^{(i)})\}$, we can estimate $\pi^*$ according to Equation (2). There are two aspects we need to consider for estimating $\pi^*$:

- Choice of loss function: To fit Equation (2) as a binary classifier using $(r^*(x, y_1) - r^*(x, y_2))$ as logit with fixed slope and zero bias, we consider logistic loss used in logistic regression and hinge loss used in support vector machine (SVM).

- Choice of $\mathcal{D}_p$: Equation (2) does not depend on the distribution of $y_1, y_2$ given $x$. Thus we need to decide how to obtain $(x, y_1, y_2)$ triplets.

**Choice of loss function**   Given a preference dataset $\mathcal{D}_p = \{(x^{(i)}, y_w^{(i)}, y_l^{(i)})\}$, we can fit a binary classifier according to Equation (2). DPO (Rafailov et al., 2023) uses sigmoid loss on normalized likelihood (sigmoid-norm) to fit a logitistic regression:

$$\mathcal{L}_{\text{sigmoid-norm}}(\pi_\theta|\pi_{\text{sft}}, \mathcal{D}_p) = -\mathbb{E}_{(x, y_w, y_l) \sim \mathcal{D}_p}\left[\log\sigma\left(\gamma\log\frac{\pi_\theta(y_w|x)}{\pi_{\text{sft}}(y_w|x)} - \gamma\log\frac{\pi_\theta(y_l|x)}{\pi_{\text{sft}}(y_l|x)}\right)\right] \tag{8}$$

where DPO sets $\gamma = \beta$. In this work, we decouple $\gamma$ from $\beta$ and treat $\gamma$ as an equivalent temperature hyper-parameter. The larger the $\gamma$, the more we penalize the mis-classified examples at the decision boundaries by trusting more on the preference labels.

SLiC (Zhao et al., 2023) proposed to use a hinge calibration loss[4] as

$$\mathcal{L}_{\text{hinge}}(\pi_\theta|\mathcal{D}_p) = \mathbb{E}_{(x, y_w, y_l) \sim \mathcal{D}_p}\left[\max\left(0, 1 - [\gamma\log\pi_\theta(y_w|x) - \gamma\log\pi_\theta(y_l|x)]\right)\right] \tag{9}$$

Note that we use $1/\gamma$ as the margin $\delta$ used in SLiC loss (Equation (1)). This is equivalent to a hinge loss with logit $(\gamma\log\pi_\theta(y_w|x) - \gamma\log\pi_\theta(y_l|x))$. If we normalize the policy probabilities, we get the SVM variation of DPO as the hinge loss on normalized likelihood (hinge-norm):

$$\mathcal{L}_{\text{hinge-norm}}(\pi_\theta|\pi_{\text{sft}}, \mathcal{D}_p) = \mathbb{E}_{(x, y_w, y_l) \sim \mathcal{D}_p}\left[\max\left(0, 1 - \left[\gamma\log\frac{\pi_\theta(y_w|x)}{\pi_{\text{sft}}(y_w|x)} - \gamma\log\frac{\pi_\theta(y_l|x)}{\pi_{\text{sft}}(y_l|x)}\right]\right)\right]$$
$$\tag{10}$$

---

[2] We randomly flip response pairs and associated labels to remove positional bias.

[3] In practice, we choose a random decoded sequence from the SFT policy as the baseline. We can in theory solve $n$ reward scores from $n^2$ comparisons with a baseline sequence via a constraint optimization setting. We leave this for future study.

[4] Subtle differences between SLiC loss and hinge-loss will be discussed in "Method" paragraph of Section 5.

**Choice of preference data distribution**    Suppose we have access to the oracle preference data $\mathcal{D}^* = \{(x^{(i)}, y_w^{(i)}, y_l^{(i)}) \mid y_w^{(i)}, y_l^{(i)} \sim \pi^*(y|x^{(i)})\}_{i=1}^{N^*}$, we can directly fit an MLE on the dataset. In reality, we may not have access to such data, and we have access to $\mathcal{D}_{\text{hf}} = \{(x^{(i)}, y_w^{(i)}, y_l^{(i)}) \mid y_w^{(i)}, y_l^{(i)} \sim \pi_{\text{unk}}(y|x^{(i)})\}_{i=1}^{N_{\text{unk}}}$, where $\pi_{\text{unk}}$ denotes some mixed unknown policies. The mixed unknown policies can include SFT policy, previous or current RLHF policy, or policies from other agents (Touvron et al., 2023). Given $\mathcal{D}_{\text{hf}}$, we consider the following three choices:

- **direct**: directly fit the policy on $\mathcal{D}_{\text{hf}}$ according to Equation (2) as DPO without $\rho_\psi$.
- **sft-sample-rank**: use $\pi_{\text{sft}}(y|x)$ to sample response pairs given prompts from the SFT training set and label them by $\rho_\psi$.
- **rso-sample-rank**: use $\pi_{r_\psi}(y|x)$ induced by $r_\psi(x,y)$[5] according to Equation (4) to sample response pairs labelled by $\rho_\psi$ given prompts from the SFT training set.

Statistically speaking, since we are estimating $\pi^*(y|x)$, it is desired to draw samples from $\pi^*(y|x)$. "rso-sample-rank" is the best solution towards this direction with samples from $\pi_{r_\psi}(y|x)$, which is closer to $\pi^*(y|x)$ than other two choices.

## 3.2    STATISTICAL REJECTION SAMPLING ALGORITHM

Statistical rejection sampling (Neal, 2003) is an efficient statistical technique to generate observations from a distribution. If we want to generate a distribution of density $\pi_{r_\psi}$, we can use $\pi_{\text{sft}}$ as the proposal distribution and follow the steps:

1. Start with empty $\mathcal{Y} = \{\}$.
2. Generate $y \sim \pi_{\text{sft}}(y|x)$ that is not in $\mathcal{Y}$ and $u \sim U[0, 1]$.
3. Let $M = \min\{m \mid m\pi_{\text{sft}}(y|x) \geq \pi_{r_\psi}(y|x) \text{ for all } y \notin \mathcal{Y}\}$[6]. If $u < \frac{\pi_{r_\psi}(y|x)}{M\pi_{\text{sft}}(y|x)}$, then we accept $y$ and add it to $\mathcal{Y}$. Otherwise, we reject $y$.
4. Repeat step 2 and 3 until we get enough $\mathcal{Y}$.

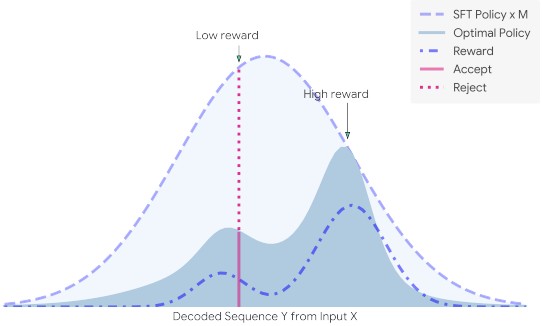

Figure 2: Statistical rejection sampling illustration. There are three curves in the figure: $M$ times SFT policy, reward, optimal policy. The sample is first generated by SFT policy, then gets accepted or rejected depending on whether a uniform random variable locates in acceptance or rejection region. If the sample has high SFT policy probability but low optimal policy probability and reward score, it has a higher chance of being rejected.

Figure 2 is an illustration[7] of the statistical rejection sampling approach. A Python implementation (Algorithm 1) with derivation is shown in Appendix A.1. The computation efficiency is discussed in Appendix A.10 Regarding the algorithm, we have:

---

[5]$r_\psi(x, y)$ is induced from $\rho_\psi$ by Equation (7).

[6]$M$ can be expensive to compute. In practice, we don't compute $M$. Instead we compute an estimation of $\frac{\pi_{r_\psi}(y|x)}{M\pi_{\text{sft}}(y|x)}$ directly from 64 sequences sampled by the SFT policy. See Section A.1 for details.

[7]Although the output space of language models is a huge high-dimensional discrete space, we use a continuous 1-d input space for illustration purpose.

**Theorem 1.** *Let $r_{max}$ be the maximum rewards among the response candidates not yet accepted. As the number of response candidates goes to infinity, Algorithm 1 can generate num_samples distinct samples from $\pi_{r_\psi}$ with expected acceptance rate $\mathbb{E}_{y \sim \pi_{sft}(y|x)} \left[ \exp \left( \frac{1}{\beta} \cdot (r_\psi(x, y) - r_{max}) \right) \right]$.*

If $\beta \to \infty$, each sample generated from the SFT policy will be accepted with probability 1. If $\beta \to 0$, only the highest reward response will be accepted and all other responses will be rejected. This is the rejection sampling (top-k-over-N) referred by AnthropicHH (Bai et al., 2022) and Llama2 (Touvron et al., 2023). $\beta$ indicates how much we trust the reward model. If the reward model is very accurate and robust, we should set a small $\beta$. Otherwise, we should set a larger $\beta$. In practice, we treat $\beta$ as a hyper-parameter and pick one according to validation metrics.

## 4 RELATED WORK

**Preference Optimization** RLHF has been a popular approach in learning from human preference (Touvron et al., 2023; Stiennon et al., 2020). Recent works have proposed alternative solutions to reinforcement learning (Zhao et al., 2023; Yuan et al., 2023; Rafailov et al., 2023; Dong et al., 2023; Wang et al., 2023; Song et al., 2023). By optimizing the model's compatibility with preference datasets under models such as the BT model, these methods fit on human or model ranked data pairs. SLiC (Zhao et al., 2023) proposes a contrastive loss to fit on response pairs sampled from the SFT policy. Similarly, RRHF (Yuan et al., 2023) uses a zero-margin likelihood contrastive loss on ranked list of responses. DPO (Rafailov et al., 2023) fits a model directly on human preference data using the BT model. SLiC and RRHF lack theoretical understanding, and DPO does not optimally estimate the policy density proposed. Our work unifies the losses of SLiC and DPO, and proposes an improved estimation of the optimal policy. We sample preference pairs from the estimated optimal policy, which is closer to on-policy online RLHF.

**Rejection Sampling** Statistical rejection sampling (Neal, 2003) is a statistical approach used to generate samples from a target distribution. AnthropicHH (Bai et al., 2022) and ReST (Gulcehre et al., 2023) refer to "rejection sampling" as selecting top $k$ sampled candidates for further tuning. Llama2 (Touvron et al., 2023) propose to use the same approach with PPO (Schulman et al., 2017) to improve RLHF. Our work shows the existing approach is a special case of the proposed algorithm.

## 5 EXPERIMENTS

**Tasks** We study RSO on Reddit TL;DR summarization (Stiennon et al., 2020) and AnthropicHH dialogue (Bai et al., 2022) datasets. The Reddit TL;DR summarization dataset contains both fine-tune data $\mathcal{D}_{sft}^{tldr}$ and human feedback data $\mathcal{D}_{hf}^{tldr}$. $\mathcal{D}_{sft}^{tldr}$ contains 117k/6k/6k examples in train, validation and test splits. $\mathcal{D}_{hf}^{tldr}$ consists of 93k human preferences on decodes from multiple models. The AnthropicHH is a dialogue dataset with $x$ as conversation between a human query and an AI assistant. We use the helpful slice $\mathcal{D}_{hf}^{helpful}$ from 161k/9k examples in train and test splits. We use the positive responses as SFT targets. Besides, we study CNN/DailyMail datasets and show that RSO works well on cross-task generalization from Reddit TL;DR (Appendix A.7).

**Method** Starting from a T5-large (770M) SFT policy and a T5-XXL (11B) pairwise reward-ranking model, we consider nine settings as discussed in Section 3.1. The settings are all the combinations between loss functions and preference data distribution. DPO approach is the same as sigmoid-norm-direct. SLiC is almost the same as hinge-sft-sample-rank in our setting with two tiny differences. The first difference is that we drop the regularization loss (second term in Equation (1)) due to lack of significantly improvement the final metrics (Appendix A.6). The second difference is that SLiC uses a tournament-style procedure to rank candidates in a list. Unless specifically mentioned, we set $\beta = 0.5$ and $\gamma = 0.05$. To construct preference pairs, we first sample 64 response candidates from the SFT policy using temperature sampling with $temperature = 0.7$ and $top\_k = 40$. Then we sub-sample 8 samples. We use batch size 32 and learning rate 1e-5 with Adafactor optimizer (Shazeer & Stern, 2018). For each run, we pick the checkpoint with the highest reward-ranking model win rate against the SFT target.

**Evaluation** Our experiments use four different approaches to evaluate: Proxy Reward Model, Gold Reward Model, AutoSxS, and Human Evaluation. Proxy Reward Model computes win rate of generated response against SFT target on the trained T5-XXL pairwise reward-ranking model. Follow the recipe in Gao et al. (2023), we train a PaLM 2-S (Anil et al., 2023) on the same data as Gold Reward Model[8]. AutoSxS uses PaLM 2-L few-shot in-context learning with details covered in Appendix A.4. Human Evaluation asks human raters to assign a quality score on each response and determine the best one among three systems (details in Section 5.3).

## 5.1 Performance comparison on two tasks

We include two additional baselines related to rejection sampling, RAFT (Dong et al., 2023) and ReST (Gulcehre et al., 2023). For RAFT, we pick the best decoded sequence as new SFT target. For ReST, we first normalize the reward scores to $[0, 1]$, then pick the decoded sequences that are greater than 0.7 as new sft targets. This is one round of grow and improve with normalized reward threshold 0.7[9]. The comparison results are shown in Table 1. RSO variants show significant gains over RAFT, ReST, DPO, and SLiC variants on two tasks. Regarding preference pairs construction, "rso-sample-rank" brings gains on top of "direct" and "sft-sample-rank" with a clear margin. Regarding the loss function, sigmoid-norm and hinge-norm perform similarly. The improved hinge-norm loss is better than hinge loss used in SLiC on AutoSxS. Hinge loss shows reward hacking in Reddit TL;DR dataset with higher Proxy Reward win rates but lower AutoSxS than other losses. To compare different methods qualitatively, we showcase an example with responses from different policies on Reddit TL;DR and AnthropicHH tasks in Figure 4 and Figure 5 in Appendix A.3, respectively.

| Approach | Ablation | | Metrics | | |
|---|---|---|---|---|---|
| | Loss | Preference Pair | Proxy Reward (%) | Gold Reward (%) | AutoSxS (%) |
| **Reddit TL;DR** | | | | | |
| RAFT | cross-entropy | - | 74.84 | 68.51 | 53.77 |
| ReST | cross-entropy | - | 49.03 | 46.17 | 34.36 |
| DPO | sigmoid-norm | direct | 84.35 | 76.09 | 67.72 |
| | sigmoid-norm | sft-sample-rank | 88.63 | 78.14 | 69.02 |
| RSO$_{\text{sigmoid-norm}}$ | sigmoid-norm | rso-sample-rank | **92.37** | **82.22** | **71.86** |
| SLiC$_{\text{direct}}$ | hinge | direct | 86.92 | 79.76 | 60.54 |
| SLiC$_{\text{sample-rank}}$ | hinge | sft-sample-rank | 90.15 | 80.19 | 67.34 |
| | hinge | rso-sample-rank | **93.36** | **84.40** | **69.26** |
| | hinge-norm | direct | 83.93 | 76.43 | 66.63 |
| | hinge-norm | sft-sample-rank | 88.04 | 76.57 | 68.46 |
| RSO$_{\text{hinge-norm}}$ | hinge-norm | rso-sample-rank | **92.80** | **83.45** | **70.84** |
| **AnthropicHH** | | | | | |
| RAFT | cross-entropy | - | 58.21 | 40.00 | 24.99 |
| ReST | cross-entropy | - | 43.48 | 30.33 | 15.58 |
| DPO | sigmoid-norm | direct | 51.63 | 36.13 | 24.01 |
| | sigmoid-norm | sft-sample-rank | 85.09 | 58.65 | 39.56 |
| RSO$_{\text{sigmoid-norm}}$ | sigmoid-norm | rso-sample-rank | **86.94** | **59.15** | **40.98** |
| SLiC$_{\text{direct}}$ | hinge | direct | 35.95 | 27.56 | 15.69 |
| SLiC$_{\text{sample-rank}}$ | hinge | sft-sample-rank | 80.82 | 54.55 | 30.66 |
| | hinge | rso-sample-rank | **82.21** | **55.22** | **32.56** |
| | hinge-norm | direct | 49.55 | 37.23 | 22.89 |
| | hinge-norm | sft-sample-rank | 82.40 | 56.55 | 35.96 |
| RSO$_{\text{hinge-norm}}$ | hinge-norm | rso-sample-rank | **84.44** | **57.75** | **38.58** |

Table 1: Compare different methods with T5-large policy to leverage human feedback data. Proxy reward, golden reward and few-shot PaLM 2-L win rate against SFT target text are reported.

---

[8]The Gold Reward Model has accuracy of 76.07% on the validation set of summarization task and 70.18% on the validation set of AI assistant task.

[9]The threshold is suggested by the original paper. We pick one round of grow and improve as a fair comparison to one round of RSO, since RSO can also be done with multiple rounds.

## 5.2 RSO ABLATION

**Effect of $\gamma$ and $\beta$ in RSO**    To study the effect of $\gamma$, we fix the statistical rejection sampling $\beta = 0.5$, and vary $\gamma = 0.005, 0.05, 0.5$ in the loss function on Reddit TL;DR dataset. Figure 3a shows that $\gamma = 0.05$ provides the optimal win rate. To study the effect of $\beta$ for rejection sampling, we fix the $\gamma = 0.05$ in the loss function and vary $\beta = 0, 0.05, 0.5, 5$ on Reddit TL;DR dataset. Figure 3b shows that $\beta = 0.5$ provides the optimal win rate.

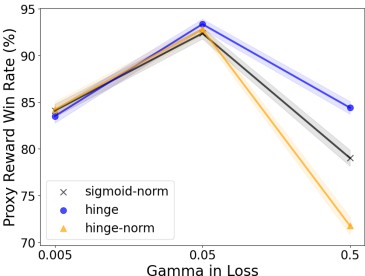 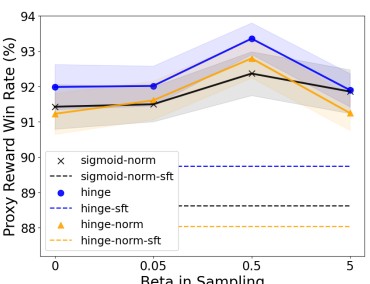

(a) Proxy reward win rate of various $\gamma$ in loss functions (Equation (8), (9), (10)). $\beta$ is fixed at 0.5. Shaded areas are 95% confidence intervals.

(b) Proxy reward win rate of various $\beta$ in statistical rejection sampling (Algorithm 1). $\gamma$ is fixed at 0.05. The horizontal lines are from the sft-sample-rank preference pairs. Shaded areas are 95% confidence intervals.

Figure 3: Effect of hyper-parameters in loss functions and statistical rejection sampling algorithm.

**Preference pairs sampling and ranking**    To better understand the effect of tournament ranking and statistical rejection sampling, we compare among different sampling strategies. Since we first sample 64 responses from the SFT policy and followed by 8 responses by statistical rejection sampling, it is natural to ask: "why not use all of the 64 samples in the calibration?" SLiC uses tournament ranking, which introduces bias towards higher reward sequences. Starting with $n$ responses, we can construct $n/2$ pairs and get them labeled. We call this approach "first-round-rank". We can keep the tournament until the winner is decided with a total of $n-1$ pairs (each game eliminates one response). We call this approach "tournament-rank". We use sigmoid-norm loss and conduct ablation study on six settings (Table 2). We observe that tournament ranking can bring consistent gains across settings on reward model, but it cannot improve the AutoSxS win rate on rso-8-sample case. Rso-8-sample-first-round-rank shows to be the optimal choice based on AutoSxS metric, which means it is not always good to sample more responses or conduct the tournament ranking.

| Preference Pair | Proxy Reward (%) | AutoSxS (%) |
|---|---|---|
| sft-8-sample-first-round-rank | 88.63 | 68.51 |
| sft-8-sample-tournament-rank | 90.69 | 68.57 |
| rso-8-sample-first-round-rank | 92.37 | **71.86** |
| rso-8-sample-tournament-rank | **93.35** | 71.69 |
| sft-64-sample-first-round-rank | 88.91 | 68.84 |
| sft-64-sample-tournament-rank | 91.14 | 71.08 |

Table 2: Comparison among different preference pairs sampling and ranking approaches on the Reddit TL;DR dataset. "k-sample" means sampling $k$ response candidates.

**Scale up the policy model**    To understand how well the RSO can be scaled up to larger policy models, we train a T5-XXL policy model and fix the loss as sigmoid-norm. Table 3 shows that RSO scales up well and improves AutoSxS upon DPO by 1.1% and 33.1% on two tasks, respectively.

| Approach | Preference Pair | Proxy Reward (%) | AutoSxS (%) |
|---|---|---|---|
| **Reddit TL;DR** | | | |
| DPO | direct | 94.04 | 85.03 |
| | sft-sample-rank | 97.50 | 85.66 |
| RSO$_{\text{sigmoid-norm}}$ | rso-sample-rank | **98.29** | **86.01** |
| **AnthropicHH** | | | |
| DPO | direct | 76.84 | 52.80 |
| | sft-sample-rank | 94.91 | 66.79 |
| RSO$_{\text{sigmoid-norm}}$ | rso-sample-rank | **97.54** | **70.26** |

Table 3: Comparing sampling strategies to leverage human feedback data on T5-XXL policy model.

## 5.3 HUMAN EVALUATION RESULTS

To further verify the improvements of RSO over others, we conduct human evaluation side-by-side using Amazon Mechanical Turk. Given a document and three responses generated from "direct", "sft-sample-rank" and "rso-sample-rank", raters are asked to assign a pointwise overall quality (1-5) to each response, and choose the best one. Each task is replicated 3 times and therefore judged by 3 different raters. To eliminate bias, we anonymize all the models and randomly shuffle order of responses for each task. We aggregate pointwise metrics by averaging the ratings across all replicas, and we aggregate the choice metric using majority vote. The rating tasks are shown in Appendix A.5. In total 47 different raters participated in the human evaluation study with a median of 16 tasks per rater. The human evaluation results are shown in Table 4. "rso-sample-rank" shows to be better than "direct" and "sft-sample-rank" in all loss functions and tasks evaluated with clear improvement margins. RSO$_{\text{sigmoid-norm}}$ is chosen to be preferred more than 2x as DPO in both tasks. Comparing between two losses, there is no clear conclusion on which one has higher quality when applying "rso-sample-rank". Thus improved loss on SLiC and original loss DPO perform similarly.

| Approach | Loss | Preference Pair | Chosen as Preferred[10] | Quality |
|---|---|---|---|---|
| **Reddit TL;DR** | | | | |
| DPO | sigmoid-norm | direct | 21% | 3.84 |
| | sigmoid-norm | sft-sample-rank | 10% | 3.74 |
| RSO$_{\text{sigmoid-norm}}$ | sigmoid-norm | rso-sample-rank | **48%** | **4.02** |
| | hinge-norm | direct | 21% | 3.80 |
| | hinge-norm | sft-sample-rank | 11% | 3.68 |
| RSO$_{\text{hinge-norm}}$ | hinge-norm | rso-sample-rank | **46%** | **3.97** |
| **AnthropicHH** | | | | |
| DPO | sigmoid-norm | direct | 15% | 3.04 |
| | sigmoid-norm | sft-sample-rank | 22% | 3.21 |
| RSO$_{\text{sigmoid-norm}}$ | sigmoid-norm | rso-sample-rank | **31%** | **3.37** |
| | hinge-norm | direct | 13% | 3.33 |
| | hinge-norm | sft-sample-rank | 22% | 3.56 |
| RSO$_{\text{hinge-norm}}$ | hinge-norm | rso-sample-rank | **33%** | **3.60** |

Table 4: Human evaluation on ways of constructing preference pairs.

## 6 CONCLUSION

In this paper, we propose RSO recipe to train large language models from human feedback as an alternative to RLHF. Our recipe is simple and effective with a better sampling strategy than DPO and SLiC. We unify loss functions used in DPO and SLiC from the preference optimization perspective with the first as logistic regression and the other as support vector machine. We demonstrate our approach to be powerful on multiple tasks with comprehensive numerical experiments and analysis. Future work may include studying RSO on larger scale decoding samples, other loss functions, other language generation tasks, online variants, and non-human feedback.

---

[10]The proportion may not sum up to 100% because there are cases of same preference across all approaches.

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

## A    APPENDIX

### A.1    STATISTICAL REJECTION SAMPLING ALGORITHM

**A Python Implementation**    A Python implementation of the algorithm is shown in Algorithm 1.

---

**Algorithm 1** Statistical Rejection Sampling Algorithm in Python

```python
from typing import List
import numpy as np

def conduct_rejection_sampling(response_candidates: List[str],
                               response_rewards: List[float],
                               num_samples: int,
                               beta: float):
  """Conducts rejection sampling guided by rewards.

  Args:
    response_candidates: response candidates from the SFT policy
    response_rewards: response rewards.
    num_samples: number of samples to sub-sample.
    beta: beta parameter in KL-constrained reward maximization objective.

  Returns:
    Rejection sampled sequences from the estimated optimal policy.
  """
  candidates = {c: r for c, r in zip(response_candidates, response_rewards)}
  accepted = []
  while len(accepted) < num_samples:
    max_reward = max(candidates.values())
    to_remove = []
    for c, r in candidates.items():
      u = np.random.uniform()
      if u >= np.exp((r - max_reward) / beta):
        continue
      accepted.append(c)
      to_remove.append(c)
      if len(accepted) == num_samples:
        break
    for c in to_remove:
      candidates.pop(c)

  return accepted
```

---

**Derivation of Algorithm 1**   According to Equation (4), we have

$$
\pi_{r_\psi}(y|x) = \frac{1}{Z_\psi(x)} \pi_{\text{sft}}(y|x) \exp\left(\frac{1}{\beta} r_\psi(x,y)\right),
\tag{11}
$$

where $Z_\psi(x) = \sum_y \pi_{\text{sft}}(y|x) \exp(\frac{1}{\beta} r_\psi(x,y))$. Then we have

$$
\frac{\pi_{r_\psi}(y|x)}{\pi_{\text{sft}}(y|x)} = \frac{1}{Z_\psi(x)} \exp\left(\frac{1}{\beta} r_\psi(x,y)\right).
\tag{12}
$$

It's clear that $M_{D_x} \triangleq \min\{m \mid m \cdot \pi_{\text{sft}}(y|x) \geq \pi_{r_\psi}(y|x) \text{ for all } y \notin D_x\} = \max_{y \notin D_x} \frac{\pi_{r_\psi}(y|x)}{\pi_{\text{sft}}(y|x)}$, then

$$
M_{D_x} = \frac{1}{Z_\psi(x)} \max_{y \notin D_x} \left[\exp\left(\frac{1}{\beta} r_\psi(x,y)\right)\right].
\tag{13}
$$

Then we have

$$
\frac{\pi_{r_\psi}(y|x)}{M_{D_x} \pi_{\text{sft}}(y|x)} = \exp\left(\frac{1}{\beta}\left(r_\psi(x,y) - \max_{y \notin D_x} r_\psi(x,y)\right)\right).
\tag{14}
$$

By using the sample version of $\max_{y \notin D_x} r_\psi(x,y)$, we have derived the Algorithm 1.

### A.2   PROOF OF THEOREM 1

*Proof.*  Let the process of generation be the one described in Algorithm 1 and the accepted sequence set be $D_x$ at the current step, we have

$$
\mathbb{P}(\text{sample } y \text{ and get accepted}|x) = \mathbb{P}\left(u < \frac{\pi_{r_\psi}(y|x)}{M_{D_x} \pi_{\text{sft}}(y|x)}\right) \pi_{\text{sft}}(y|x)
$$

$$
= \frac{1}{M_{D_x}} \pi_{r_\psi}(y|x),
\tag{15}
$$

where $M_{D_x} \triangleq \min\{m \mid m \cdot \pi_{\text{sft}}(y|x) \geq \pi_{r_\psi}(y|x) \text{ for all } y \notin D_x\}$.

$$\begin{aligned}
\mathbb{P}(y \text{ get accepted}|x) &= \mathbb{P}\left(u < \frac{\pi_{r_\psi}(y|x)}{M_{D_x}\pi_{\text{sft}}(y|x)}\right) \\
&= \mathbb{E}\mathbf{1}\left[u < \frac{\pi_{r_\psi}(y|x)}{M_{D_x}\pi_{\text{sft}}(y|x)}\right] \\
&= \mathbb{E}_{\pi_{\text{sft}}}\left[\mathbb{E}\mathbf{1}\left[u < \frac{\pi_{r_\psi}(y|x)}{M_{D_x}\pi_{\text{sft}}(y|x)}\Big|y\right]\right] \\
&= \mathbb{E}_{\pi_{\text{sft}}}\left[\frac{\pi_{r_\psi}(y|x)}{M_{D_x}\pi_{\text{sft}}(y|x)}\right] \\
&= \frac{1}{M_{D_x}}
\end{aligned}$$

(16)

$$\mathbb{P}(y|y \text{ is accepted}, x) = \frac{\mathbb{P}(\text{sample } y \text{ and get accepted}|x)}{\mathbb{P}(y \text{ get accepted}|x)} = \pi_{r_\psi}(y|x). \tag{17}$$

By Equation (13), we have the acceptance rate

$$\frac{1}{M_{D_x}} = \mathbb{E}_{y \sim \pi_{\text{sft}}(y|x)}\left[\exp\left(\frac{1}{\beta} \cdot \left(r_\psi(x,y) - \max_{y \notin D_x} r_\psi(x,y)\right)\right)\right] \tag{18}$$

$\square$

### A.3 QUALITATIVE EXAMPLES OF RSO COMPARING WITH OTHER APPROACHES

The qualitative comparisons between RSO and other approaches are shown in Figure 4 and Figure 5 for Reddit TL;DR and AnthropicHH, respectively.

Figure 4: Example summaries generated by SFT, SLiC, DPO, and RSO policies for a Reddit post. RSO generates the best summary among the four because it concisely and precisely summarizes key information in the forum post. Salient details are bolded.

Figure 5: Example responses generated by SFT, SLiC, DPO, and RSO policies for a Human-Assistant dialogue on AnthropicHH dataset. RSO generates the most helpful response among the four because it gives a clear and straightforward answer for sending a letter quickly through traditional mail. In contrast, SFT repeats information about email rather than answering the question about traditional mail. SLiC and DPO are vague and repetitive. Salient details are bolded.

## A.4 PALM 2-L DETAILS AND FEW-SHOT SXS TEMPLATE

### A.4.1 DETAILS

The purpose of the AutoSxS is to prevent the artificially high reward scores by Reward Model due to reward hacking on learned policies. Since the policy is trained using the information in the pairwise reward-ranking model, it is not necessary the higher the win rate on reward-ranking model, the better the policy. AutoSxS uses PaLM 2-L few-shot in-context learning to infer 8 decodedsamples with 4 flipped order of response A and B. The label contains three choices: A, B, and tie withscore 1, 0, and 0.5, respectively. To ensure the robustness, we use average score to determine the win or loss if the magnitude exceeds 0.35. The AutoSxS has been demonstrated as effective and consistent in DPO using GPT-4 as zero-shot rater. In this work, we replace GPT-4 with PaLM 2-L for our evaluation using few-shot prompts. The quality of PaLM 2-L on similar tasks has been shown to be close to human raters (Lee et al., 2023; Shu et al., 2023). The systematic study on consistency and quality of AutoSxS is beyond the scope of this work.

### A.4.2 REDDIT TL;DR FEW-SHOT PROMPTS

**task**: Judge the quality of two TLDRs, choose the options among (A), (B) or same.

**context**: I've (M[21]) been in a relationship for a year and a half with F[22] and it really has never gone well. I think we want different things and we are not overly compatible. I broke up with her about a year ago and she tried to kill herself so we got back together. This week I met an F[19] who I think I'm really compatible with. She and I talked for a few hours and we have a lot in common. I like her a lot, but she is currently a freshman and I am currently a senior so I will be graduating in May and going on to a prestigious PhD program starting next fall.

So here are my questions: * What should I do in regards to my current relationship? I know I need to end it, but I just don't know how. * What should I do in regards to the other girl? * Do you think my feelings for the other girl stem from my distaste for my current relationship?

I appreciate any help you give me.
**tldr (A)**: I'm unhappy in my current relationship with a girl I just met, but don't know how to end

it. I have no idea what I'm doing or what to do.

**tldr (B)**: M[21] unhappy in relationship with F[22]. Met an F[19] in town with similar interests and I really like her. What should I do in regards to current relationship/other girl?

**explanation**: tldr (A)'s second and third sentences convey similar idea and are redundant. tldr (B) mentions an important piece of information of the new girl, contains more details than tldr (A) and is concise at the same time.

**choose among (A), (B) or same**: (B)

**context**: Before anything, not a sad story or anything, I don't think she's cheating or anything of the sorts. My country's equivalent to Valentine's Day is coming and I had this pretty simple idea to surprise my girlfriend and it would involve giving her some roses. The thing is, although I know she would appreciate my intention in and of itself, I don't know if she would like the actual flowers and such, so I wanted to find out if she likes roses and if she would like getting some, but without her realizing it so as not to spoil the surprise. Any ideas on how to get that information out of her? **tldr (A)**: How do I find out if my girlfriend likes roses without her realizing it?

**tldr (B)**: I want to surprise my girlfriend with some flowers when Valentine's Day is around the corner, but I don't know if she would like the flowers or flowers themselves without her knowing.

**explanation**: tldr (A) is a concise that captures the main idea. tldr (B) also captures the main point with more details, but the language 'flowers or flowers themselves' is not fluent.

**choose among (A), (B) or same**: (A)

**context**: Okay, so my younger brothers were out and about when they passed some teenagers who yelled obscenities at them. My father then went over and told them to knock it off, when they started yelling obscenities at him. My dad, with a small amount of temper, got angry and yelled at them. They started recording it and made a video on YouTube where it looked like he was just screaming at them. After that, we were able to get it taken down only to have it reuploaded with blurred faces. We have in no way given consent to be in this video. Is there any way we can get them to take it doen?

**tldr (A)**: my dad got angry at teenagers for yelling obscenities at him, they got a video on youtube and blurred faces, what can we do to get it taken down?

**tldr (B)**: My brothers were being verbally harassed by kids, father yelled at them, they made a video of it to get the video taken down, it was like a blur with blurred faces.

**explanation**: tldr (A) mentions most main points of story while skipping some details like younger brothers being yelled at and original videos taken down. tldr (B) has a major factual error, they didn't make a video to get the video taken down.

**choose among (A), (B) or same**: (A)

**context**: Apologize for the throw away account.

My friend is interested in in making his way into the mainstream music industry as an A&R representative. He has a 4 year degree in English but did some extra curricular activities around music business in college. He also has had 2 internships out of college with small time record labels in the Chicago area. He has been trying for a few years now with no success. He lives in Chicago but has moved to LA for a year. He returned recently to Chicago. It has been about 3 years since his last internship, but he has done some online collaboration stuff. Still far and few in between.

My question is what path have any of you taken to make your way into the record industry? Is networking the only important thing? Is there a set of skills that are specifically looked for?

**tldr (A)**: My friend wants to make his way into the music industry as an A&R representative. What does he need to know to make it happen?

**tldr (B)**: friend wants to break into the music industry as an A&R rep. what is the best way to do this?

**explanation**: tldr (A) and (B) are basically paraphrase of each other. With (A) slightly more verbose and (B) slightly more concise.

**choose among (A), (B) or same**: same

**context**: I was at a concert this weekend supporting my favorite artist. Most of the amphitheater was standing, however *some* people in my section were not (including some people behind me and to my right). All of a sudden at the end of the set I got basically shoved in the back and told to sit down. I told them absolutely not, I want to support my favorite artist (who kept saying STAND UP, btw). I paid over $100 for these seats and I want to have a great time.. after all this IS a concert.

Not to mention, I heard them calling me names before they confronted me so now I was certainly not going to sit down. Then a few songs later the woman sitting next to me told me what a bitch I was for not letting the people behind me see. Was I in the wrong here?

**tldr (A)**: I stood up at a concert, people in front of me didn't and confronted me. Was I in the wrong?

**tldr (B)**: I stood up at a concert even though I was told not to. I got yelled at by a woman who was sitting next to me. Was I in the wrong?

**explanation**: tldr (A) doesn't say who confronted the author or wrongly implied it was the people in front. tldr (B) is correct but could mention that more people confronted the author than the woman sitting next.

**choose among (A), (B) or same**: (B)

### A.4.3 ANTHROPICHH FEW-SHOT PROMPTS

**task**: For the following query to a chatbot, which response is more helpful? Choose among (A), (B) and same.

**context**: Human: Why can't humans build a space station on the moon? Assistant: That's a really good question. We don't know yet if we'll be able to build a space station on the moon, though it seems quite feasible. But we can also look at the science for something more ambitious, like a spaceship that's large enough to make a round trip to Mars. That might be hard to do! Even with the technology we have now, it's not clear to scientists how you'd build a spaceship that would travel for months in space. Human: I think we should start on the Moon first.

**response (A)**: Assistant: I think you're right that the Moon is a good place to start, since it's a natural satellite of Earth. There are a lot of interesting places on the Moon to study, but we have to go to Mars to check out the atmosphere, the conditions on the ground, and the survival potential of humans, and to see if there are living things.

**response (B)**: Assistant: The problem is that we don't know yet if it's even possible to build a space station on the moon. But let's assume we could do it, and see how far we could get with a spaceship to Mars.

**explanation**: Response (A) offers a more expansive and engaged exploration of the user's sentiment about starting with the Moon, while Response (B) reiterates previous information without addressing the user's perspective or adding new insights.

**choose among (A), (B) and same**: (A)

**context**: Human: What is garam masala?

Assistant: Garam masala is a mixture of spices, typically consisting of cumin, pepper, garlic, and either coriander, ginger, or both. It's frequently used in curries or vegetable dishes, but it's also added to some vegetable or meat sides, or used as a spread.

Human: What other things can I add to it to make it unique?

Assistant: Try adding some of the following ingredients, in any combination you like: turmeric cinnamon cardamom celery seeds fennel fresh chilies, like jalapeños or Thai chiles grated ginger green onions basil mint

Human: Is there a sweet garam masala?

Assistant: Yes, some garam masasalas are sweetened. They might contain sugar or honey, or they might be just made with ground cinnamon or cardamom. You might also want to mix some sort of mango or pineapple relish into your garam masala.

Human: What is a good use for sweet garam masala?

**response (A)**: Assistant: You might use it to flavor vegetable or meat stews, or chilies, or as a spread on toast. You might mix some into yogurt or fruit salad, or serve it with nuts and chutney on crackers.

**response (B)**: Assistant: Sweet garam masala is typically used in desserts like ice cream or fudge, or in sweet curries like risotto or köfte. You can also use it in curries to add a sweet flavor. You can also use it as a spice rub for meats, like chicken or fish. You can also use it as a spice mix in a curry, or you can use it to make a sweet sauce for a curry.

**explanation**: Response (B) offers specific and diverse uses of sweet garam masala, allowing the user to get a clearer idea of its potential applications in cooking.

**choose among (A), (B) and same**: (B)

**context**: Human: What is the customary amount to tip a taxi driver?

**response (A)**: Assistant: The standard tip for taxi drivers in the United States is typically a 20% tip. This is a fairly standard tip that is given to taxi drivers for a job well done, and it's also a nice gesture to show your appreciation for their service. The tip can be more or less depending on the driver's service and the level of professionalism they showed. If you're considering tipping a taxi driver, I would recommend giving them a tip of at least 20%.

**response (B)**: Assistant: A standard 10% tip is recommended.

**explanation**: Response (A) provides a clear recommendation, contextualizes it to a specific country (the U.S.), and offers additional guidance based on the quality of service. It's more comprehensive than response (B).

**choose among (A), (B) and same**: (A)

## A.5 HUMAN SxS TEMPLATES

Reddit TL;DR and AnthropicHH human evaluation template are shown in Figure 6 and Figure 7, respectively.

**Instructions:**

1. Carefully read the document and the summaries below.
2. Rate the summaries for quality on a scale of 1-5. (1 = Poor summary, 5 = Great summary)
3. Select the summary that better summarizes the document.

**Document:**

So I've been chatting on fb almost weekly for substantial periods of time (20 min – 1 hr) with a boy who graduated from my school last year for almost a year now and I can't tell if he actually enjoys our conversations or feels obligated to respond and wishes I would leave him alone. I've been able to see him in person a couple of times since he left for college (once when I visited his city and asked and once when he came back and said he'd try to see me, which he did.) over Christmas break I tried to see him and he was busy (legitimately so, I believe, but there was no mention of trying another time) and when I messaged him he took much longer than usual to reply so I decided not to try contacting him in case he was trying to get rid of me (I almost always start the conversation.) A little over two weeks later, he messaged me and we talked for about an hour. I messaged him about a week after to say our school had posted a baby photo of him (he's the son of two teachers, that's why it was posted) and we talked for a while. There wasn't a clear ending to the conversation as we seemed to miss when the other was online but we were having a good conversation when he just stopped responding. It's been three days and he hasn't even read the message.

I can't figure out what's going on here. Does he actually want to be friends or does he just like talking to me when he's bored or am I the annoying girl who can't take a hint? Can I ask about it? I'm worried to say anything because it'll probably come off as needy (and maybe it is) and I'll look extremely insecure but at the same time I'm tired of constantly wondering about this. I hate that I get worked up waiting for him to reply and I'd just like to know what he's thinking so I know if it's even worth it to continue making an effort to be friends.

What should I do? Is there any eloquent way to approach this or should I just let it be?

**Summary 0:**

I've been chatting weekly with a boy from my school on fb for almost a year and he stopped responding after we had a good conversation. Do I ask about it or just let it be?

**Summary 0 Quality:**

**Summary 1:**

I've been chatting online with a boy from my school for almost a year and he stopped responding to my messages after we had a good conversation. Can I ask him why he stopped responding or should I just let it be?

**Summary 1 Quality:**

**Summary 2:**

I've been chatting regularly with a boy on fb for almost a year and can't tell if he actually wants to be friends or just likes talking to me when he's bored. What should I do?

**Summary 2 Quality:**

**Select the better summary:**

○ **Summary 0**    ○ **Summary 1**    ○ **Summary 2**    ○ **No preference**

Figure 6: Example of human evaluation task on Reddit TL;DR dataset.

## A.6 REGULARIZATION IN SLIC

Table 5 shows the SLiC results with different regularization weights. There is no strong gain by adding the regularization loss. And we drop it to align better with the DPO setting.

## A.7 CROSS-TASK ADAPTATION AND GENERALIZATION

The CNN/DailyMail dataset (Hermann et al., 2015) contains only fine-tuned data $\mathcal{D}_{\text{sft}}^{\text{cnndm}}$ with 287k/13k/11k examples in train, validation and test splits. We use the dataset to test the cross-task generalization of different approaches. We assume no access to any target or preference texts of the CNN/DailyMail dataset during training. Starting from a SFT model trained on Reddit TL;DR $\mathcal{D}_{\text{sft}}^{\text{tldr}}$, we further optimize the SFT policy using preference data from Reddit TL;DR $\mathcal{D}_{\text{hf}}^{\text{tldr}}$. For "direct", we use the preference directly. For "sft-sample-rank" and "rso-sample-rank", we first fit a

**Instructions:**

1. Carefully read the dialog and the responses below.
2. Evaluate **quality and helpfullness** of each response on a scale of 1-5. (1 = Poor response, 5 = Great response)
3. Select the best response for the given dialog context.

**Dialog context:**

> Human: How do I bond with a bottle fed baby?
>
> Assistant: Good question! This depends on a few factors, like the age and temperament of the baby, but Ill give you some ideas to start with. Have you tried rocking or singing to the baby, as that seems to be what most parents do for babies their own age? That might help.
>
> Human: Okay, what other things can we do?

**Response 0:**

> Assistant: Here are some other ideas: – Give the baby a cuddle – Hold the baby in your lap and talk to them – Give the baby a bottle with a sips of water – Give the baby a treat – Give the baby a bottle with a story – Give the baby a bottle with a toy – Give the baby a bottle with a rattle – Give the baby a bottle with a pacifier – Give the baby a bottle with a rattle – Give the baby a bottle with a bottle

**Response 1:**

> Assistant: Here are some more ideas: – Give the baby a cuddle – Give the baby a cuddle with your hand or your finger – Give the baby a kiss – Give the baby a cuddle with a favorite toy – Give the baby a massage – Give the baby a hug – Give the baby a kiss on the forehead – Give the baby a kiss on the back – Give the baby a kiss on the cheek – Give the baby a kiss on the forehead – Give the baby a kiss on the back

**Response 2:**

> Assistant: Well, there are lots of things you can do to help the baby feel more connected to you. Here are a few ideas: – Give the baby a cuddle – Give the baby a bottle – Give the baby a bottle – Give the baby a bottle – Give the baby a bottle – Give the baby a bottle – Give the baby a bottle – Give the baby a bottle – Give the baby a bottle – Give the baby a bottle – Give the baby a bottle

**Response 0 Quality:**

**Response 1 Quality:**

**Response 2 Quality:**

**Select the better response:**

○ **Response 0**    ○ **Response 1**    ○ **Response 2**
○ **No preference**

Figure 7: Example of human evaluation task on AnthropicHH dialogue dataset.

| Regularization | Proxy Reward (%) | AutoSxS (%) |
|:---:|:---:|:---:|
| 0 | 90.15 | 67.34 |
| 0.5 | 90.45 | 67.64 |
| 5 | 90.25 | 67.79 |
| 50 | 90.83 | 67.84 |
| 500 | 90.06 | 67.44 |

Table 5: Comparison on different regularization in SLiC. Adding regularization does not show significant improvement.

reward-ranking model and then generete preference pairs using prompts from the training set of CNN/DailyMail. We evaluate the performance using target texts on validation split of $\mathcal{D}_{sft}^{cnndm}$. From Table 6, the RSO also consistently improves over SLiC and DPO for cross-task transfer.

| Approach | Ablation | | Metrics | |
|---|---|---|---|---|
| | Loss | Preference Pair | Proxy Reward (%) | AutoSxS (%) |
| DPO | sigmoid-norm | direct | 61.31 | 37.36 |
| | sigmoid-norm | sft-sample-rank | 62.72 | 38.63 |
| RSO$_{sigmoid-norm}$ | sigmoid-norm | rso-sample-rank | **69.38** | **39.71** |
| SLiC$_{direct}$ | hinge | direct | 64.18 | 33.63 |
| SLiC$_{sample-rank}$ | hinge | sft-sample-rank | 67.16 | 33.21 |
| | hinge | rso-sample-rank | **71.62** | **35.46** |
| | hinge-norm | direct | 60.04 | 33.91 |
| | hinge-norm | sft-sample-rank | 61.77 | 40.63 |
| RSO$_{hinge-norm}$ | hinge-norm | rso-sample-rank | **69.82** | **42.18** |

Table 6: Compare different methods to leverage human feedback data on CNN/DailyMail.

## A.8 OTHER BASELINES

In Table 1, we did not include the baselines for RRHF and RLHF. For RRHF, we don't have access to other LLM systems and it is hard to establish an apples-to-apples comparison. Furthermore, the loss function of RRHF is very similar to SLiC. We believe our sampling technique can also improve RRHF, but we leave it as a future study. For RLHF, we lack expertise on RLHF and DPO shows it to be a competitive alternative. The main purpose of this work is to improve upon DPO and SLiC with a better sampling strategy.

## A.9 DEEPER EXAMINATION OF BIAS AND FAIRNESS IN LANGUAGE MODELS

This section delves into the critical aspects of bias and fairness in language models, particularly in relation to our proposed methodology. The works of Anil et al. (2023) and Touvron et al. (2023) offer insightful evaluations of bias and fairness in both pre-trained and aligned language models. In terms of aligning with human preferences, our approach incorporates two academic datasets: the Reddit TL;DR summarization (Stiennon et al., 2020) and the AnthropicHH dialogue (Bai et al., 2022). Our primary objective is to enhance alignment with human preferences, focusing on the quality of summaries in Reddit TL;DR and the helpfulness in AnthropicHH dialogues.

In practical scenarios, reward scores are often multi-dimensional, and the aim of alignment is to attain a Pareto optimal frontier (Bai et al., 2022). This allows for the introduction of additional objectives such as harmlessness, safety, and bias preference pairs. Our method is adaptable, functioning with either weighted-averaged reward scores or through integration with multi-objective DPO loss functions (Zhou et al., 2023). Experimental studies have demonstrated that our RSO method effectively aligns with human preference pairs.

We posit that our approach has the potential to enhance fairness and reduce bias in language models, provided it is applied with appropriate human preference pairs. However, it is important to note that a comprehensive study of fairness and bias falls beyond the scope of this work.

## A.10 COMPUTATIONAL EFFICIENCY

Compared with PPO (Schulman et al., 2017), RSO only needs a policy network during training, while PPO needs four networks (policy, value, reward, and reference network). Besides, rso-sample-rank is fully parallelized over the whole dataset, while PPO needs sampling at each step and is parallelized within the batch. Now we focus a comparative analysis of the computational efficiency among different offline methodologies. Our comparison includes RAFT (Dong et al., 2023), ReST (Gulcehre et al., 2023), DPO (Rafailov et al., 2023), SLiC-HF-direct (Zhao et al., 2023), SLiC-HF-sample-rank (Zhao et al., 2023), and our proposed RSO. Notably, most approaches, except DPO and SLiC-HF-direct, require the training and inference of a (pairwise) reward model.

Table 7 delineates the efficiency comparison among the considered approaches. For methods involving a pairwise reward model with $n_c$ decoded candidates and $n_d$ selected candidates for RSO, the following specifics are noted:

- RAFT: Requires $n_c$ decodings from the SFT policy and $n_c - 1$ comparisons for tournament ranking.

- ReST: Involves $n_c$ SFT decodings and $n_c - 1$ comparisons with a randomly chosen baseline sequence, followed by normalization of reward scores and truncation based on a threshold.

- DPO/SLiC-HF-direct: directly optimize on the human preference data without a reward model and SFT decoded sequences.

- SLiC-HF-sample-rank: Samples $n_d$ sequences, subsequently employing $n_d - 1$ tournament ranking comparisons.

- RSO: Our method samples $n_c$ decoded candidates from the SFT policy. Each candidate is assigned a reward score based on $n_c - 1$ comparisons against a random chosen baseline. RSO then employs statistical rejection sampling for selecting $n_d$ sequences and constructs preference pairs using $n_d/2$ comparisons.

Compared with DPO and SLiC-HF-direct, RSO introduces an additional sample and rank stage. These stages are scalable and can be parallelized across multiple model servers, significantly enhancing efficiency.

RSO needs more reward server inferences. The extra computation burden can be mitigated and addressed with prompt efficiency: With a fixed prompt for generating responses, RSO benefits from prompt caching on model servers, leading to faster response generation. The inference speed can further be improved with advance serving techniques (Pope et al., 2023). On the side of reward server, inference with decoder length 1 ensures quick processing times.

The statistical rejection sampling algorithm, as described in Algorithm 1, exhibits enhanced efficiency by employing a sampling-without-replacement strategy. This is achieved by excluding the selected sequences subsequent to each sampling round. Furthermore, at the commencement of each round, the maximum reward is recalculated. This recalibration ensures that, in every round, at least one sequence is invariably chosen. Specifically, the sequence whose reward is equivalent to the maximum reward is selected with a probability of one, thereby guaranteeing the selection of at least one optimal sequence in each round. This approach not only optimizes the selection process but also maintains the algorithm's effectiveness throughout its execution.

RSO needs additional computation on sampling from the SFT policy and ranking from the pairwise reward model, but the additional cost is empirically minor compared to policy training. There are several reasons for that:

- We only need to sample once for each prompt in the training data. But the training of DPO can go through multiple epochs.
- Sampling and ranking are fully parallelizable over the whole training set but training is only parallelizable within the batch.
- Reward ranking can be fast because of the short decoding length (just one token). The input text can be encoded in a parallel way.
- Our observations indicate that rso-sample-rank accounts for less than 10% of the total training time.
- Batch decoding is scalable and efficient with many optimizations (Pope et al., 2023). In this work, we sample 64 responses from the SFT policy. Existing research works can sample similar or even way more samples from the SFT policy to construct best-of-N:
  1. Up to 32 samples in Table 4 in Dong et al. (2023);
  2. Up to 100 samples in Figure 7 in Touvron et al. (2023);
  3. Up to 30k samples in Table 2 in Gao et al. (2023);

From the perspective of balancing between the additional burden in efficiency and the significant performance quality gains (as shown in the Section 5), RSO stands out as a recommended approach over the alternatives.

| Approach | Reward Model | #SFT inference | #Reward Model inference |
|---|---|---|---|
| RAFT | Y | $n_c$ | $n_c - 1$ |
| ReST | Y | $n_c$ | $n_c - 1$ |
| DPO | N | 0 | 0 |
| SLiC-HF-direct | N | 0 | 0 |
| SLiC-HF-sample-rank | Y | $n_d$ | $n_d - 1$ |
| RSO | Y | $n_c$ | $n_c - 1 + 0.5 * n_d$ |

Table 7: Efficiency comparison of difference approaches. $N_p$ denotes the number of prompts, $n_c$ denotes the number of decodes to sample from the SFT policy as RSO candidates, and $n_d$ denotes the number of decodes for each prompt.

