# OpenReview forum: "Statistical Rejection Sampling Improves Preference Optimization"
_ICLR.cc/2024/Conference — ICLR 2024 poster_

### Official Review · Reviewer_nTnq · 2023-10-28

**Soundness:** 3 good
**Presentation:** 4 excellent
**Contribution:** 3 good
**Rating:** 6
**Confidence:** 3

**Summary:**

The paper introduces an innovative method, Statistical Rejection Sampling Optimization (RSO), for RLHF. RSO employs rejection sampling to extract preference data from the optimal target policy, facilitating a more precise estimation of this policy. Additionally, this paper provides a unified framework for previous approaches, i.e., SLiC and DPO. Experimental outcomes indicate that RSO consistently surpasses existing methods, as evaluated by both large language models and human raters.

**Strengths:**

1. Presents a novel approach for RLHF using rejection sampling.
2. Unifies prior work, including SLiC and DPO, within a comprehensive review.
3. Provides comprehensive and robust experiments with SOTA, demonstrating clear enhancements of RSO.

**Weaknesses:**

1. The unifying link between DPO and SLiC appears to be deliberately designed, particularly noticeable in the normalization term, i.e., $\pi_{sft}(y|x)$, in Eq 10.
2. The theoretical conclusion that the policy induced from RSO in Eq 4 is optimal seems questionable. RSO continues to use the proxy reward model to label responses, regardless of whether they're from the pre-trained or learned policy. However, the reward model is learned from $D_p$ and not $D_p^*$ without further updates, which could introduce approximation errors and bias in subsequent labeling and learning steps. In the other word, the policy induced from Eq 4 is not optimal because of the non-optimal reward model.
3. Works with a similar focus on rejection sampling, such as ReST(http://arxiv.org/abs/2308.08998), ALMoST (https://arxiv.org/abs/2305.13735) RAFT (https://arxiv.org/abs/2304.06767), should be involved in comparison.

**Questions:**

Why do not directly sample responses from $\pi_{r_{\psi}}(y|x)$ (with a reward threshold like ReST) rather than sampling from $\pi_{sft}(y|x)$ with the statistical rejection sampling to approximate the samples from  $\pi_{r_{\psi}}(y|x)$?

---

> ### Author Response · Authors · 2023-11-13
>
> >Re1: The unifying link between DPO and SLiC appears to be deliberately designed, particularly noticeable in the normalization term, i.e., $\pi_\text{sft}(y|x)$, in Eq 10.
>
> Answer: Your observation about the unifying link between DPO and SLiC in the context of Eq 10 is indeed insightful. However, I believe there are a few additional points to consider, which might offer a different perspective.
>
> 1. The purpose of integrating $\pi_\text{sft}(y|x)$ into Eq 10 goes beyond merely unifying DPO and SLiC. This integration is primarily aimed at establishing a more robust theoretical framework for the SLiC-HF approach. While it's true that SLiC-HF and DPO emerged around the same time, leading to some confusion about their connection, their starting points are fundamentally different.
>
>     * SLiC-HF is grounded in the concept of likelihood calibration, where a better response is expected to have a higher likelihood than a less desirable response, by a defined margin.
>
>     * On the other hand, DPO builds on the Reinforcement Learning from Human Feedback (RLHF) objective and the Bradley-Terry model, leading to the derivation of the sigmoid loss function as a Maximum Likelihood Estimation (MLE).
>
> 2. Our contribution, as outlined in the paper, is to theoretically link these two approaches. By substituting the Bradley-Terry model with a Support Vector Machine (SVM) model, we obtain the hinge-norm of SLiC, effectively a normalized version. The original SLiC calibration loss overlooks the SFT policy normalization, relying solely on preference pairs labeled by the reward model. The hinge-norm loss, conversely, uses the SFT likelihood as a baseline to compute the relative likelihood ratio between the current policy and the SFT policy. This approach not only calibrates the likelihood ratio towards human preferences but also acts as an adaptive margin version of the SLiC loss. The margin, in this case, is influenced by the SFT likelihoods of the better and worse responses. Furthermore, this paper [1] highlights issues with the Bradley-Terry model and proposes the IPO loss, which aligns more closely with the hinge-norm loss than the original SLiC.
>
> 3. In terms of implementation, DPO and SLiC variants share similarities by the choice of link functions and normalization. There isn't conclusive evidence yet to prefer one over the other. The paper provides a comprehensive framework for choosing between these options, underscoring the need for further research in this area.
>
> In conclusion, while the connection between DPO and SLiC might seem deliberate, especially in the context of Eq 10, it is part of a broader effort to establish a more theoretically sound approach to SLiC-HF, taking into account the nuances of different models and the importance of the SFT policy.
>
>
> **References**
>
> [1] Azar, Mohammad Gheshlaghi, et al. "A general theoretical paradigm to understand learning from human preferences." arXiv preprint arXiv:2310.12036 (2023).

---

> ### Author Response · Authors · 2023-11-13
>
> >Re2: The theoretical conclusion that the policy induced from RSO in Eq 4 is optimal seems questionable. RSO continues to use the proxy reward model to label responses, regardless of whether they're from the pre-trained or learned policy. However, the reward model is learned from $D_p$ and not $D_p^∗$ without further updates, which could introduce approximation errors and bias in subsequent labeling and learning steps. In the other word, the policy induced from Eq 4 is not optimal because of the non-optimal reward model.
>
> Answer: Thank you for pointing out this important issue. We agree that RSO in Eq 4 is not strictly optimal. We have addressed the questions one-by-one.
>
> (1) “RSO continues to use the proxy reward model to label responses, regardless of whether they're from the pre-trained or learned policy.”
>
> In our revised approach, we acknowledge the impracticality of directly sampling from an optimal policy. To address this, we've shifted from using the term 'optimal policy' to 'estimated optimal policy' in our documentation, specifically when referencing the sourcing of preference pairs from the rejection sampled policy. This change underlines our strategy to approximate the optimal policy via a rejection sampling method. The distribution shift of reward model training data also exists in RLHF, even though it conducts sampling from online policy. The errors in the proxy reward model can result in reward hacking. To mitigate the issue, we can collect preference data from the current best policy iteratively. In Llama2 [1], they collect human preference data from the current best policy on a weekly basis, as the policy keeps improving. Our work can follow the same approach as we improve the policy iteratively using rso-sample-rank.
>
> (2) “However, the reward model is learned from $D_p$ and not $D_p^*$ without further updates, which could introduce approximation errors and bias in subsequent labeling and learning steps”
>
> We acknowledge the concerns regarding approximation errors and bias in our reward model learned from $D_p$. However, we assert that our pairwise reward model holds significant advantages over the implicit pointwise reward model (Equation (5)) induced by the Bradley-Terry model as per DPO, for several reasons:
>
> 1. **Pointwise vs Pairwise**: Our pairwise comparison approach, structured as “[Context] {context} [Response A] {response a} [Response B] {response b}”, aligns more naturally with human cognitive processes, compared with assigning a real-valued reward score for a response given a context.  The pairwise approach not only simplifies the task but also leverages the extensive knowledge transfer from pre-trained language models in classification and comparison tasks. Unlike the Bradley-Terry model, which imposes a rank-1 approximation thus potentially losing information, our model directly represents the complexity of pairwise interactions without such assumptions.
>
> 2. **Implicit vs Explicit**: Our model is an explicit text-to-text classification model, a task that is straightforward and less prone to the complexities associated with language generation. In contrast, the implicit reward model in DPO, based on likelihood scoring or generation, navigates a more intricate language output space. This complexity can hinder the effective transfer of knowledge from pre-trained models. Our approach, grounded in the principle that classification tasks are inherently more manageable than generation tasks, offers a more robust and reliable framework.
>
> To substantiate our claims, our comprehensive experimental studies demonstrate the superiority of our model. For instance, rso-sample-rank shows to be significantly better than DPO. These results highlight not just the theoretical but also the practical advantages of our approach.
>
>
> **References**
>
> [1] Touvron, Hugo, et al. "Llama 2: Open foundation and fine-tuned chat models." arXiv preprint arXiv:2307.09288 (2023).

---

> ### Author Response · Authors · 2023-11-13
>
> >Re3: Works with a similar focus on rejection sampling, such as ReST(http://arxiv.org/abs/2308.08998), ALMoST (https://arxiv.org/abs/2305.13735) RAFT (https://arxiv.org/abs/2304.06767), should be involved in comparison.
>
> Answer: Thank you for your suggestions regarding our baseline comparisons. We acknowledge the significance of ALMoST in the context of iterative LLMs and PPO algorithms. However, we opted not to include it in our baseline due to its distinct approach, particularly its reliance on synthetic preference pairs, which diverges significantly from RSO's methodology. This fundamental difference in approach could lead to an unfair comparison.
>
> In contrast, RAFT and ReST share more similarities with RSO in terms of objectives and methodologies. For RAFT, we continue to fit the SFT model using cross-entropy loss on the best response selected by tournament ranking using a pairwise reward model. For ReST, we fix $\tau=0.7, I=1, G=1$ as suggested by the ReST paper. We normalize the reward scores between 0 and 1 and choose the responses with reward greater than 0.7 as new SFT targets. We did not include multiple rounds of grow and improve stages because of a fair comparison consideration. RSO can also be done in multiple rounds to generate better rso-sample-rank preference pairs.
>
> Our results demonstrate that RSO outperforms these robust baselines across various metrics, highlighting its effectiveness. While we are confident in our findings, we remain open to future studies that may explore these comparisons further, potentially integrating models like ALMoST under a framework that allows for a fair comparison.
>
> | Task          | Approach           | Proxy Reward (%) | Gold Reward (%) | AutoSxS (%) |
> | ------------- | ------------------ | ---------------- | --------------- | ----------- |
> | Reddit TL; DR | RAFT               | 74.84            | 68.51           | 53.77       |
> |               | ReST               | 49.03            | 46.17           | 34.36       |
> |               | $RSO_{sigmoid-norm}$ | 92.37            | 82.22           | **71.86**       |
> |               | $RSO_{hinge-norm}$   | **92.80**           | **83.45**           | 70.84       |
> | AnthropicHH   | RAFT               | 58.21            | 40.00           | 24.99       |
> |               | ReST               | 43.48            | 30.33           | 15.58       |
> |               | $RSO_{sigmoid-norm}$ | **86.94**            | **59.15**           | **40.98**       |
> |               | $RSO_{hinge-norm}$   | 84.44            | 57.75           | 38.58       |

---

> ### Author Response · Authors · 2023-11-13
>
> >Re4: Why do not directly sample responses from $\pi_{r_\psi}(y|x)$ (with a reward threshold like ReST) rather than sampling from $\pi_\text{sft}(y|x)$ with the statistical rejection sampling to approximate the samples from $\pi_{r_\psi}(y|x)$?
>
> Answer: Thank you for raising this important question. There are two reasons that we don’t directly sample responses from $\pi_{r_\psi}(y|x)$:
>
> 1. **Online vs Offline**: ReST is an online on-policy algorithm, while RSO is an offline algorithm. Online algorithm allows the model to sample responses from the current policy, while the offline algorithm only has access to the initial policy (SFT policy) to generate responses. RSO bridges the gap by sampling responses from a better policy that is closer to the optimal policy. Online algorithms like ReST and PPO are not time and memory efficient. ReST needs policy network and value network to co-exist during training, and sampling from the online policy is not parallelizable. PPO needs two more networks (reference policy network and value network). On the other hand, RSO can generate rso-sample-rank preference pairs from SFT policy with a rejection sampling algorithm in a fully parallelizable way. The training of the RSO policy also only needs the policy network, which is much more memory efficient than ReST/PPO.
>
> 2. **Connection with ReST**: in our paper, we demonstrate that the prevalent best-of-N or top-k-over-N rejection sampling methods are merely specific instances of our more comprehensive statistical rejection sampling algorithm, particularly when the beta parameter is set to zero. The ReST paper is very similar to an online version of the top-k-over-N. In Figure 3(b), we show RSO is better than the best-k-over-N. We also verify that in our updated draft in comparison with ReST. If beta approaches infinity, our method aligns seamlessly with sampling from an SFT policy. This crucial insight underpins our proposed method, which adeptly balances reliance on the SFT policy with the accuracy of the reward model. We use the hyper-parameter beta to control the balance between the two and thus offer a more robust and adaptable solution for practical applications.

---

> ### Comment · Reviewer_nTnq · 2023-11-20
> **Official Response from Reviewer nTnq**
>
> The reviewer thanks the authors for their prompt response.
> > R1: The unifying link between DPO and SLiC appears to be deliberately designed.
>
> The reason why I say it was deliberately designed is that I find there is a regularization term in Eq. (1) (the second term). However, I also find the statement from the authors: "The ﬁrst difference is that we drop the regularization loss (second term in Equation (1)) due to lack of signiﬁcantly improvement the ﬁnal metrics (Appendix A.6).". I do not 100% buy this statement because the second term is important to avoid the model collapse. More, the ablation study of Table 1 (Zhao et al., 2023) shows further performance improvement with best-ranked candidates in the regularization term.
>
> > R3: Experiments comparison between more benchmarks.
>
> The experiments demonstrate that RSO significantly outperforms (x1.8) ReST! Could the author provide further clarification for this? Further, I think ReST can also use the sft-sample-rank and rso-sample-rank for further improvement. Please correct me if I'm wrong somewhere.
>
> > Re4: Why do not directly sample responses from $\pi_{r_\psi}(y \mid x)$ (with a reward threshold like ReST) rather than sampling from $\pi_{\mathrm{sft}}(y \mid x)$ with the statistical rejection sampling to approximate the samples from $\pi_{r_\psi}(y \mid x)$?
>
> 1. online vs. offline: RSO is somewhat of an offline algorithm. However, we only call it offline when we never sample any response from any policy. As a result, the memory efficiency is not valid for me as RSO still samples actions from the behavior policy.
> 2. Based on the statement "The ReST paper is very similar to an online version of the top-k-over-N.", I can not observe the clear superiority of RSO from the methodology perspective, as well as the memory view, compared with ReST. Besides, this "offline" approximation brings some disadvantages such as statistical approximation, and computation cost. Regarding "balances reliance on the SFT policy with the accuracy of the reward model.", why should use the balance word because the reward model never changes? Suppose I have a misunderstanding with the reward model, I can simply set a relaxed threshold in ReST to "balance" these two.

---

> > ### Author Response · Authors · 2023-11-20
> >
> > >Re1: The reason why I say it was deliberately designed is that I find there is a regularization term in Eq. (1) (the second term). However, I also find the statement from the authors: "The ﬁrst difference is that we drop the regularization loss (second term in Equation (1)) due to lack of signiﬁcantly improvement the ﬁnal metrics (Appendix A.6).". I do not 100% buy this statement because the second term is important to avoid the model collapse. More, the ablation study of Table 1 (Zhao et al., 2023) shows further performance improvement with best-ranked candidates in the regularization term.
> >
> > Answer (1/2): Thanks for further clarification on this. We agree that it is important to clarify the difference between the SLiC loss and the hinge-loss. By making that clear, the unified view between DPO and SLiC is more smooth and natural.
> >
> > (1) “The reason why I say it was deliberately designed is that I find there is a regularization term in Eq. (1) (the second term).”
> >
> > This is a great point. We should have pointed out the missing regularization term when we first introduce the choice of loss functions at Section 3.1. In our revised manuscript, we have replaced “SLiC (Zhao et al., 2023) proposed to use a hinge loss” with “SLiC (Zhao et al., 2023) proposed to use a hinge calibration loss”. We have also  added a footnote to clarify the difference between original SLiC loss and hinge-loss at Section 3.1.
> >
> > (2) “However, I also find the statement from the authors: "The ﬁrst difference is that we drop the regularization loss (second term in Equation (1)) due to lack of signiﬁcantly improvement the ﬁnal metrics (Appendix A.6).". I do not 100% buy this statement because the second term is important to avoid the model collapse.”
> >
> > In response to the skepticism about dropping the regularization term, we provide a detailed rationale supported by both theoretical and empirical evidence.
> >
> > 1. **Offline Setting Justification**: Our model operates in an offline setting, exclusively using responses sampled from the SFT policy. This inherently limits deviation from the SFT policy, as demonstrated in [1]. The probability of significant deviation is further quantified by the KL divergence formula presented in Appendix G.3 of [2].
> >
> > 2. **SLiC Loss Regularization Choices**: In the SLiC loss framework (Equation (4) in [3]), we explore two $y_\text{ref}$ options: SFT target and the best candidate ranked by the reward model. Each has distinct implications for the need for regularization:
> >
> >     * **SFT Target**:
> >
> >         1) **Redundancy of Additional Information**: The SFT target information is already incorporated into the SFT policy. Therefore, further training with the regularization term may not contribute additional insights. SLiC's approach, akin to DPO, adjusts the probability distribution between responses but maintains stability without explicit regularization.
> >
> >         2) **Implicit Regularization via Calibration Term**: The SLiC margin ($\delta$) acts as an inherent control mechanism, paralleling the role of the KL regularization in the DPO model. It ensures that the learned policy remains aligned with the SFT policy, as demonstrated by varying margin values.
> >
> >         3) **Empirical Validation**: Appendix A.6 and Figures in our paper provide numerical evidence against significant performance degradation or model collapse when excluding the regularization term. This practical observation aligns with our theoretical expectations.
> >
> >     * **Best Candidate**:
> >
> >         1) **Gradient Equivalence**: if we use the best candidate in the regularization term, the gradient of the log-likelihood on that candidate will be the same as the one inside the calibration loss when we use that candidate as the positive response (Equation (4) in [3]), as long as the calibration loss is non-zero.
> >
> >         2) **Tournament Ranking Weightage**: In SLiC, the best candidate inherently gains sufficient weightage through tournament ranking. This setup ensures that the calibration loss already incorporates the necessary adjustments to the likelihood of the best candidate, making additional regularization less impactful.
> >
> > Our comprehensive approach, combining theoretical understanding and empirical validation, affirms that excluding the regularization term does not adversely affect our model's stability or performance in the specific offline context. This decision is in line with the foundational principles of the DPO model and is supported by our data-driven analyses.
> >
> > **References**
> >
> > [1] Gao, Leo, John Schulman, and Jacob Hilton. "Scaling laws for reward model overoptimization." International Conference on Machine Learning. PMLR, 2023.
> >
> > [2] Stiennon, Nisan, et al. "Learning to summarize with human feedback." Advances in Neural Information Processing Systems 33 (2020): 3008-3021.
> >
> > [3] Zhao, Yao, et al. "Slic-hf: Sequence likelihood calibration with human feedback." arXiv preprint arXiv:2305.10425 (2023).

---

> > ### Author Response · Authors · 2023-11-20
> >
> > > Re1: The reason why I say it was deliberately designed is that I find there is a regularization term in Eq. (1) (the second term). However, I also find the statement from the authors: "The ﬁrst difference is that we drop the regularization loss (second term in Equation (1)) due to lack of signiﬁcantly improvement the ﬁnal metrics (Appendix A.6).". I do not 100% buy this statement because the second term is important to avoid the model collapse. More, the ablation study of Table 1 (Zhao et al., 2023) shows further performance improvement with best-ranked candidates in the regularization term.
> >
> > Answer (2/2):
> >
> > (3) “More, the ablation study of Table 1 (Zhao et al., 2023) shows further performance improvement with best-ranked candidates in the regularization term.”
> >
> > In the ablation study of Table 1 in [3], it shows the further fine-tuning on the best-ranked candidates can improve the performance upon SFT policy. That study has no calibration loss. With the calibration loss, the SLiC paper [3] does not conduct a systematic study on the effectiveness of the regularization term at different regularization levels. This work conducts the systematic ablations and shows that the regularization term may not be critical (see the above paragraphs for explanations) once we have the calibration loss. To also ensure a fair comparison between SLiC and DPO, we decided to remove the term.
> >
> > **References**
> >
> > [1] Gao, Leo, John Schulman, and Jacob Hilton. "Scaling laws for reward model overoptimization." International Conference on Machine Learning. PMLR, 2023.
> >
> > [2] Stiennon, Nisan, et al. "Learning to summarize with human feedback." Advances in Neural Information Processing Systems 33 (2020): 3008-3021.
> >
> > [3] Zhao, Yao, et al. "Slic-hf: Sequence likelihood calibration with human feedback." arXiv preprint arXiv:2305.10425 (2023).

---

> > ### Author Response · Authors · 2023-11-20
> >
> > > Re2: The experiments demonstrate that RSO significantly outperforms (x1.8) ReST! Could the author provide further clarification for this? Further, I think ReST can also use the sft-sample-rank and rso-sample-rank for further improvement. Please correct me if I'm wrong somewhere.
> >
> > Answer: Thank you for your insightful query regarding our experimental setup, particularly in comparing RSO with ReST. I’d like to clarify our methodology and the rationale behind it.
> >
> > (1) “The experiments demonstrate that RSO significantly outperforms (x1.8) ReST! Could the author provide further clarification for this?”
> >
> > We use the setting of $\tau=0.7,I=1,G=1$, as suggested by the ReST paper (Section 4 in [1]). It means one round of “grow” and one round of “improve” with normalized reward threshold 0.7. We clarify why we choose this setting:
> >
> >   1. **Why $G=1$**: this is for fair comparison. Each round of “grow” needs a new set of sampled and scored responses. For DPO/SLiC/RSO, we just conduct one round of sampling and scoring. If we go beyond one round of growth, sft-sample-rank and rso-sample-rank can be repeated with the new policy learned.
> >
> >   2. **Why $I=1$**: we can observe diminishing returns of increasing number of rounds for “improve” (Figure 3 and Figure 4(b) in [1]). The majority gains are shown from $I=0$ to $I=1$. Meanwhile, RAFT [2] can be considered as a special case of ReST with $\tau$ as 1. In that case, all different $I$ will result in the same result. In our draft, we show that RAFT is no better than RSO. Given limited time and resources during rebuttal, we did not further investigate more “improve rounds”. We admit that with “I” becoming larger, we may expect further improvement on ReST. Correspondingly, $\tau$ captures how much to trust the reward model, which is somewhat related to $\gamma$ in our RSO loss. A more fair comparison with multiple rounds of “improve” should be compared with RSO with dynamically annealed $\gamma$. We may systematically study the proper “I” and update the draft with the best “I” in our camera-ready version.
> >
> >   3. **Why $\tau=0.7$**: this is mentioned by the original ReST paper (Section 4 in [1]) that they choose thresholds from a sequence of increasing values [0.0, 0.7, 0.8, 0.9, 0.95, 0.99]. We used the recommended value from the ReST paper.
> >
> > (2) “Further, I think ReST can also use the sft-sample-rank and rso-sample-rank for further improvement. Please correct me if I'm wrong somewhere.”
> >
> > We indeed use the responses sampled from SFT policy. For rso-sample-rank, 64 responses are from SFT policy and statistical rejection sampling algorithm is applied on filtering 8 responses among all those candidates. For ReST, we start with the 64 responses sampled from the SFT policy, and normalize the reward values of them between 0 and 1. Then we apply $\tau=0.7$ to collect the high-reward responses for ReST training.
> >
> > **References**
> >
> > [1] Gulcehre, Caglar, et al. "Reinforced self-training (rest) for language modeling." arXiv preprint arXiv:2308.08998 (2023).
> >
> > [2] Dong, Hanze, et al. "Raft: Reward ranked finetuning for generative foundation model alignment." arXiv preprint arXiv:2304.06767 (2023).

---

> > ### Author Response · Authors · 2023-11-20
> >
> > > Re3.1: online vs. offline: RSO is somewhat of an offline algorithm. However, we only call it offline when we never sample any response from any policy. As a result, the memory efficiency is not valid for me as RSO still samples actions from the behavior policy.
> >
> > Answer: thanks for raising these points for further clarification.
> >
> > 1. **RSO is strictly an offline algorithm**: Offline RL refers to the scenario where an RL algorithm learns a policy from a static, pre-collected dataset. RSO uses the data sampled from a static SFT policy, followed by rejection sampling based on a static reward model. The dataset is indeed pre-collected and static before the policy learning. Thus RSO is a strictly offline algorithm.
> >
> > 2. **Memory efficiency**:  RSO is both computation and memory more efficient than online algorithms such as PPO. RSO **NEVER** samples examples from the current policy. It only uses the samples from the SFT policy in a smarter way. It sub-selects samples among the SFT responses such that the selected ones follow a distribution that is closer to the optimal policy. We always use only one network in memory across all stages.
> >
> >   * **SFT sample**: during the SFT sample stage, we only load SFT policy for decoding (associated with the log-likelihood scores), which is scalable with parallel decoding over the whole training set.
> >
> >   * **Reward ranking**: during the reward model ranking stage, we only load Proxy Reward Model for ranking, which is also scalable with parallel ranking over the whole training set.
> >
> >   * **Policy learning**: during the preference optimization training stage, we only load the policy in memory. We can read the reference log-likelihood from the rso-sample-ranked dataset, because during the SFT sample stage, the log-likelihood scores are written in the decoding sequences as an additional field.
> >
> > So unlike online algorithms such as PPO, RSO gets rid of the value network, reward model, and SFT policy during the training stage. It reduces memory usage from $4p$ to $p$ (assuming each network consumes $p$ parameters). Furthermore, RSO can conduct parallel decoding, rewarding, and input encoding caching. PPO do not have those scalability properties. More discussions on efficiency can be found at Table 5 in [1].
> >
> > **References**
> >
> > [1] Zhao, Yao, et al. "Slic-hf: Sequence likelihood calibration with human feedback." arXiv preprint arXiv:2305.10425 (2023).

---

> > ### Author Response · Authors · 2023-11-20
> >
> > > Re3.2: Based on the statement "The ReST paper is very similar to an online version of the top-k-over-N.", I can not observe the clear superiority of RSO from the methodology perspective, as well as the memory view, compared with ReST. Besides, this "offline" approximation brings some disadvantages such as statistical approximation, and computation cost. Regarding "balances reliance on the SFT policy with the accuracy of the reward model.", why should use the balance word because the reward model never changes? Suppose I have a misunderstanding with the reward model, I can simply set a relaxed threshold in ReST to "balance" these two.
> >
> > Answer (1/2): Thank you for asking for further clarification.
> >
> > (1) ““Based on the statement "The ReST paper is very similar to an online version of the top-k-over-N.", I can not observe the clear superiority of RSO from the methodology perspective, as well as the memory view, compared with ReST.”
> >
> > By stating that "The ReST paper is very similar to an online version of the top-k-over-N.", we refer to the iterative stage with multiple rounds of “grow” stage. From the memory view, we agree that ReST and RSO always use one network in memory since all the training data are pre-collected.
> >
> > From methodology perspective, ReST uses cross-entropy loss as behavior cloning, while RSO uses contrastive loss like those in DPO and SLiC. We believe that the RSO has advantages over ReST based on the following reasons:
> >
> >   1. **Theoretical foundation**: starting from the RLHF objective and Bradley-Terry model, we can derive the MLE as a contrastive loss such as sigmoid-norm (DPO) and hinge-norm (normalized SLiC). ReST lacks such theoretical understanding and guarantees. ReST emphasizes that the high reward sequence has high likelihood, which is to estimate the “mode” of a distribution. On contract, RSO tunes the relative difference between positive and negative responses, which is a more fine-grained adjustment of the shape of the distribution around a high probability region. That is why we treat it as a likelihood calibration approach.
> >
> >   2. **Elegance and computational efficiency**: RSO is more elegant and computationally efficient.
> >
> >       * **Fewer hyper-parameters**: In RSO, we only have two hyper-parameters $\beta$ and $\gamma$, while in ReST, it has many hyper-parameters, including number of “grow” $G$, number of improvement $I$, and a sequence of thresholds $\[\tau_0,\tau_1,...,\tau_I]$. The hyper-parameter choice can be very computationally expensive.
> >
> >       * **One round vs. multiple-rounds**: RSO just conducts one round of policy learning, while ReST needs to conduct multiple-rounds of “growth” and “improvement”. Each round of “improve” needs to monitor the validation loss and change the training datasets. Each round of “grow” needs to re-sample the responses from the current policy.
> >
> >   3. **Empirical evidence**: there are a few findings to support RSO being superior to ReST.
> >
> >       * Zephyr [1] found that using DPO loss can result in a better student policy.
> >
> >       * SLiC-HF [2] found that calibration loss can improve upon the Best-of-N fine-tuning.
> >
> >       * Our new results show that RSO is better than RAFT and one round of ReST with $I=1,G=1,\tau=0.7$.
> >
> > (2) "Besides, this "offline" approximation brings some disadvantages such as statistical approximation, and computation cost."
> >
> > 1. **Statistical approximation**: We agree that “offline” approximation brings disadvantages in statistical approximation. We argue that the pairwise reward model is more accurate and powerful than the implicit reward model. Thus the approximation is towards a better direction compared to DPO. ReST also faces statistical approximation issues such as the choice of thresholds. An inaccurate proxy reward model will also cause errors in selecting the training data. Furthermore, ReST does not use a pairwise reward model, which may have larger estimation errors.
> >
> > 2. **Computation cost**: the computational cost for RSO and one-round ReST ($I=1,G=1$) need the same number of decodes. Assume we sample 64 candidates and RSO uses 8 pairs, ReST needs 63 reward model inferences for one prompt and RSO needs 67, which is a small extra computational cost.
> >
> > Compared with the online approach, RSO can conduct parallel decoding, rewarding, and input encoding caching. Compared with $k$-grow-round ReST, RSO is $k$ times more computational efficient.
> >
> > **References**
> >
> > [1] Tunstall, Lewis, et al. "Zephyr: Direct Distillation of LM Alignment." arXiv preprint arXiv:2310.16944 (2023).
> >
> > [2] Zhao, Yao, et al. "Slic-hf: Sequence likelihood calibration with human feedback." arXiv preprint arXiv:2305.10425 (2023).

---

> > ### Author Response · Authors · 2023-11-20
> >
> > > Re3.2: Based on the statement "The ReST paper is very similar to an online version of the top-k-over-N.", I can not observe the clear superiority of RSO from the methodology perspective, as well as the memory view, compared with ReST. Besides, this "offline" approximation brings some disadvantages such as statistical approximation, and computation cost. Regarding "balances reliance on the SFT policy with the accuracy of the reward model.", why should use the balance word because the reward model never changes? Suppose I have a misunderstanding with the reward model, I can simply set a relaxed threshold in ReST to "balance" these two.
> >
> > Answer (2/2):
> >
> > (1) "Regarding "balances reliance on the SFT policy with the accuracy of the reward model.", why should use the balance word because the reward model never changes? Suppose I have a misunderstanding with the reward model, I can simply set a relaxed threshold in ReST to "balance" these two.”"
> >
> > By the word “balance”, we refer to the quality of SFT policy versus the quality of the proxy reward model. A high quality SFT policy means the one that can generate high quality response from any given prompt measured by the Gold Reward. A high quality proxy reward model means the one that has high validation accuracy and correlation with the Gold Reward. If the SFT policy is much better than proxy reward, we should assign a large $\beta$ in rejection sampling to prevent reward hacking towards the proxy reward model. If the proxy reward is much better than the SFT policy, we should assign a small $\beta$ in rejection sampling to encourage the policy to go far from the SFT policy towards a higher Gold Reward value. We agree that in ReST, we can also tune the thresholds to achieve a good balance between the SFT and proxy reward. But the hyper-parameter tuning can be very ineffective for multi-rounds “improve” ReST, because it requires the choice of a sequence of thresholds.

---

> ### Comment · Reviewer_nTnq · 2023-11-21
> **Official Response from Reviewer nTnq**
>
> Thanks for the further clarification. However, I still have some questions but first with the most important one:
>
> > Offline vs. online: RSO uses the data sampled from a static SFT policy, followed by rejection sampling based on a static reward model.
>
> Does RSO sample responses from $\pi_{sft}$ during the training? I think it is from Section 3.2. If it is, then RSO is not offline and the memory benefit is not valid.

---

> > ### Author Response · Authors · 2023-11-21
> >
> > >Re1: Does RSO sample responses from $\pi_\text{sft}$ during the training? I think it is from Section 3.2. If it is, then RSO is not offline and the memory benefit is not valid.
> >
> > Answer: Thanks again for raising this and we agree it appears to be confusing. Let me clarify the whole process of RSO as illustrated in Figure 1. The whole pipeline consists of three sequential and standalone steps:
> >
> > 1. **Sample**: this step only involves **inferences** from the SFT policy and the reward model.
> >
> > 2. **Rank**: this step only involves **inferences** from the reward model.
> >
> > 3. **Optimize**: the real policy training **starts at this step** with the training code and computation **identical to DPO**. The purpose of the first two steps is to pre-compute the training data for this step. This step happens after the sample and rank steps. Regarding the training process, the **only difference** between DPO and RSO is **the choice of training data**:
> >
> >     * DPO uses the reward model training data **directly**.
> >     * RSO uses the training data **pre-computed** from the step 1 and 2
> >
> > The policy training **only happens at the step 3** using the pre-collected data from the first two steps. So RSO is strictly an offline algorithm with the **SAME** memory usage as DPO at the training stage.

---

> ### Comment · Reviewer_nTnq · 2023-11-22
> **Regarding Offline vs. Online Further Question**
>
> Thanks for the further clarification. To the best of my understanding, the response from the authors seems like steps 1 and 2 are totally decoupled with step 3 (optimization). Yes, I would like to agree with the statement. However, when I back to Section 3.2:
>
> 1. empty y
> 2. generate samples
> 3. rank
> 4. repeat steps 2 and 3
>
> 5. optimize
>
> Is it a one-step optimization, regarding the learned policy $\pi_r$? In other words, Is $\pi_r$ never changed in Statistical Rejection Sampling? Could the authors provide a rough pseudocode including key steps for the whole training process to make it more clear?

---

> > ### Author Response · Authors · 2023-11-22
> >
> > > Re1: Thanks for the further clarification. To the best of my understanding, the response from the authors seems like steps 1 and 2 are totally decoupled with step 3 (optimization). Yes, I would like to agree with the statement. However, when I back to Section 3.2: 1. empty y; 2. generate samples; 3. rank; 4. repeat steps 2 and 3; 5. optimize
> >
> > Answer: Thanks for following up on this. Rejection sampling **ONLY** happens at step 1 (Sample) in Figure 1. There is no policy optimization in the rejection sampling stage. We **DON'T** have **"5. optimize"** in Section 3.2. Section 3.2 is to explain the ideas of rejection sampling, which happens at step 1 in Figure 1.
> >
> > > Re2: Is it a one-step optimization, regarding the learned policy $\pi_r$? In other words, Is $\pi_r$ never changed in Statistical Rejection Sampling?
> >
> > Answer: We would like to make it clear that $\pi_r$ is the target optimal policy that is analytically solved. We never have access to the policy in any form of neural networks. The goal of rejection sampling is to collect preference pairs that approximately follow the distribution of $\pi_r$. We don't need to have access to the policy $\pi_r$. Instead, we can generated responses that follows $\pi_r$ distribution from SFT policy with rejection sampling (Theorem 1).
> >
> > **No policy update** is involved anywhere in the rejection sampling stage. We provide a python implementation of rejection sampling in Algorithm 1 in Appendix A.1. From the Algorithm 1, you can see, it takes **only** the inputs of response candidates (generated by SFT policy) and the corresponding reward scores (generated by proxy reward).
> >
> > $\pi_r$ is the **theoretical optimal policy** in mathematical format. Thus it is never changed in any step.
> >
> >
> > > Re3: Could the authors provide a rough pseudocode including key steps for the whole training process to make it more clear?
> >
> > Answer: Our pipeline is illustrated in Figure 1. We first do a sample (step 1) and rank (step 2)  to prepare the preference pairs. After that, we use the DPO/SLiC training code for preference optimization. The training process **ONLY** happens at step 3 (Optimize). The model takes a preference pair as input with the DPO (sigmoid-norm) or hinge-norm loss.  The policy optimization is exactly **IDENTICAL** to DPO. The whole pipeline is described in the last paragraph of Section 1 ("In this work, we address the above issues by ...").
> >
> > Feel free to let us know if anything is not clear and we are more than happy to address that.

---

> > > ### Comment · Reviewer_nTnq · 2023-11-22
> > > **Official Response from Reviewer nTnq**
> > >
> > > Thanks for the throughout and patient clarification. Based on the strong experiments, responses, and revised paper, I decided to raise my score.

---

> > > > ### Author Response · Authors · 2023-11-22
> > > >
> > > > Thanks a lot for your engagement and helpful comments! The feedbacks from the reviewer and other peers are very valuable for improving the paper quality and clarity. Really appreciate your time and effort.

---

### Official Review · Reviewer_DcKN · 2023-11-01

**Soundness:** 2 fair
**Presentation:** 3 good
**Contribution:** 3 good
**Rating:** 6
**Confidence:** 4

**Summary:**

This paper proposes a novel approach called Statistical Rejection Sampling Optimization (RSO) to improve preference optimization in language models. The authors address the limitations of existing methods by introducing rejection sampling to source preference data from the optimal policy. They also propose a unified framework that enhances the loss functions used in both Sequence Likelihood Calibration (SLiC) and Direct Preference Optimization (DPO). Through extensive experiments, the authors demonstrate that RSO consistently outperforms SLiC and DPO in terms of both Large Language Models (LLMs) and human raters' evaluations.

**Strengths:**

The authors provide a well-structured review of relevant literature, highlighting the limitations of existing methods and positioning their work in the context of prior research.

**Weaknesses:**

1. The core idea of utilizing rejection sampling to filter the response feels somewhat trivial

2. Why is there a significant improvement in rso-sample-rank compared to sft-sample-rank on Reddit TL;DR, while the improvement is not as noticeable on AnthropicHH? What could be the potential reasons behind this?

3. There might be flaws in the design of the experimental part: the reward for training and testing is the same. Although there are evaluations from GPT/human, generally speaking, these are not very convincing.  You can refer to [1, 2] for Synthetic Data Setup, Recalibration, gold reward, and proxy reward to have a fair comparison.

[1] Let’s Verify Step by Step

[2] Scaling Laws for Reward Model Overoptimization

**Questions:**

Please refer to the weakness section

---

> ### Author Response · Authors · 2023-11-13
>
> > Re1: The core idea of utilizing rejection sampling to filter the response feels somewhat trivial
>
> Answer: Your feedback about the perceived triviality of using rejection sampling in our approach is appreciated. We understand that at first glance, rejection sampling might seem simplistic. However, the key contribution of our work lies in the context of its application and the nuanced improvements we have implemented.
> 1. **Bridging a Crucial Gap**: Our primary aim is to bridge the gap between offline off-policy algorithms and approximately on-policy algorithms. Current approaches like DPO and SLiC-HF have not effectively addressed the distribution-shift issue. Our work is pioneering in conducting a systematic study to approximate the optimal policy among offline human preference alignment algorithms.
> 2. **Connect with Existing Best-of-N**: In our paper, we demonstrate that the prevalent best-of-N or top-k-over-N rejection sampling methods are merely specific instances of our more comprehensive statistical rejection sampling algorithm, particularly when the beta parameter is set to zero. More importantly, we establish that as beta approaches infinity, our method aligns seamlessly with sampling from an SFT policy. This crucial insight underpins our proposed method, which adeptly balances reliance on the SFT policy with the accuracy of the reward model. Unlike existing methods, ours provides a nuanced approach that adapts dynamically to varying degrees of confidence in either the SFT policy or the reward model, thereby offering a more robust and adaptable solution for practical applications.
> 3. **Efficiency Enhancements**: We acknowledge the concerns raised by reviewer hBgS regarding the computational expense of calculating $M$. To address this, we have innovated by not computing
> $M$ directly. Instead, we estimate $\frac{\pi_{r_{\psi}}(y|x)}{M\pi_{\text{sft}}(y|x)}$ in its entirety using 64 sequences sampled through the SFT policy. This methodological change, detailed in Appendix A.1, significantly enhances efficiency.
> 4. **Improved Rejection Sampling Method**: Our method diverges from traditional statistical rejection sampling by sampling from the proposal distribution without replacement. This approach is more efficient in high-dimensional settings where the probability of rejecting many proposals is high. By recalculating the maximum reward at each round’s start among unselected candidates, we ensure the selection of at least one candidate—the one with the highest reward. This modification guarantees efficiency, particularly when the target is a limited number of selected responses.
> 5. **Theoretical Backing and Simplicity**: Our algorithm, while simple in its execution as demonstrated in Algorithm 1 in Appendix A.1, is grounded in robust theoretical principles. The calculation of the threshold for a uniform random sample is simplified to computing exp((reward-max_reward) / beta), a result of deriving a closed-form solution for the optimal policy.
>
> In summary, our approach, although founded on a basic concept like rejection sampling, is a novel and effective solution for the challenges in offline human preference alignment algorithms. The improvements we have made are not just in terms of algorithmic efficiency, but also in bridging a significant research gap. The simplicity of the method should not undermine its effectiveness and the novelty of its application in this context.

---

> ### Author Response · Authors · 2023-11-13
>
> > Re2: Why is there a significant improvement in rso-sample-rank compared to sft-sample-rank on Reddit TL;DR, while the improvement is not as noticeable on AnthropicHH? What could be the potential reasons behind this?
>
> Answer: We appreciate the reviewer's insightful observation regarding the varying degrees of improvement in rso-sample-rank compared to sft-sample-rank across the Reddit TL;DR and AnthropicHH datasets. Our analysis suggests two primary reasons for this phenomenon:
>
> 1. **Dataset Characteristics and Their Impact**:
>
>     * **Reddit TL;DR Dataset**: This dataset uniquely combines SFT and human preference data sources. The human preference responses, generated through a mix of policies from OpenAI, including best-of-N rejection sampling, often yield higher quality outputs than SFT targets. Existing literature provides evidence on this. For example, in the right panel of Figure 2 in the DPO paper [1], it shows that Prefered-FT (SFT on the better responses from human preference data) is generally better than SFT. This divergence in quality potentially enlarges the gap between the SFT policy and the optimal policy, thereby making the improvements in rso-sample-rank more pronounced.
>
>     * **AnthropicHH Dataset**: Contrarily, in the AnthropicHH dataset, both the SFT targets and reward model are derived exclusively from human preference data. The proximity in quality between the SFT policy and the reward model narrows the gap with the optimal policy, resulting in less noticeable improvements in rso-sample-rank.
>
> 2. **Evaluation Methodology**:
>     * Our human evaluation methodology, which deviates from the standard side-by-side comparison, might contribute to these observations. We asked raters to choose the best response among direct, sft-sample-rank, and rso-sample-rank options. While this approach has its merits, it may not fully capture the nuanced differences between sft-sample-rank and rso-sample-rank. Future studies could benefit from a more conventional comparative analysis to validate these findings further.
>
> **References**
>
> [1] Rafailov, Rafael, et al. "Direct preference optimization: Your language model is secretly a reward model." arXiv preprint arXiv:2305.18290 (2023).

---

> ### Author Response · Authors · 2023-11-13
>
> > Re3: There might be flaws in the design of the experimental part: the reward for training and testing is the same. Although there are evaluations from GPT/human, generally speaking, these are not very convincing. You can refer to [1, 2] for Synthetic Data Setup, Recalibration, gold reward, and proxy reward to have a fair comparison.
>
> Answer: Thank you for emphasizing the need for a robust evaluation framework in our experimental design. Your suggestions regarding the utilization of distinct rewards for training and testing phases, as well as the incorporation of a gold reward model, are indeed pertinent.
>
> In response, we have revised our approach following the methodologies outlined in [1]. Specifically, we have now implemented larger reward models (PaLM-2 S) to serve as our Gold Reward model. This model is not only used as an additional metric but also helps in differentiating the rewards between the training and testing phases, thereby addressing your concern about the potential overlap in rewards.
>
> Moreover, to ensure transparency and thoroughness in our evaluation, we have included detailed metrics on the evaluation accuracy of the Gold Reward model in Section 5 of our updated draft. This additional layer of evaluation not only bolsters the validity of our findings but also aligns our methodology more closely with current best practices in the field.
> We believe these modifications greatly enhance the rigor of our experimental setup and appreciate your valuable input in guiding these improvements.
>
> **References**
>
> [1] Gao, Leo, John Schulman, and Jacob Hilton. "Scaling laws for reward model overoptimization." International Conference on Machine Learning. PMLR, 2023.

---

> ### Comment · Reviewer_DcKN · 2023-11-21
>
> Thank you for your response and for adding a golden reward, which I believe is a crucial element.
>
> Like reviewer hBgS, I am also interested in the cost aspect of the method. I have two questions:
>
> 1. What is the value of 'num_samples'? To obtain this number of samples, how many additional samples need to be generated?
>
> 2. What is the specific training cost (GPU hours) for generating new samples plus the 'best of 64' to estimate M, and how long does it take to train DPO? I am aware that DPO trains quickly, so I am concerned about the potential excess time introduced by additional training.

---

> > ### Author Response · Authors · 2023-11-21
> >
> > > Re1: What is the value of 'num_samples'? To obtain this number of samples, how many additional samples need to be generated?
> >
> > The number of candidates and responses can be tunable. In our experiments, for each response, we sample 64 response candidates from the SFT policy and use statistical rejection sampling to sample 8 responses. Compared with DPO, we need an extra 64 decodes. Compared with sft-sample-rank, we need additional 64-8=56 decodes. Compared with RAFT/ReST, we don’t need any extra decodes because they also use all 64 candidates to compute best-of-N and the ones with high rewards.
> > In practice, batch decoding is efficient, scalable, and cost effective [1][2][3]. To further verify this, we conduct the following speed benchmarking on Llama2 7b q5-k-m with a single NVIDIA GeForce RTX 4090 GPU card with llama.cpp [4]:
> >
> > | Input length | Decode length | Number of decodes | Number of decoding tokens per second | Memory overhead | Model memory |
> > | ------------ | ------------- | ----------------- | ------------------------------------ | --------------- | ------------ |
> > | 1024         | 128           | 64                | 1135                                 | 5.7G            | 4.5G         |
> > | 1024         | 128           | 16                | 596                                  | 1.9G            | 4.5G         |
> > | 1024         | 128           | 8                 | 507                                  | 1.2G            | 4.5G         |
> > | 1024         | 128           | 1                 | 120                                  | 0.7G            | 4.5G         |
> >
> > In addition, the inference is scalable to the number of accelerators, because the parallelism can be applied across the whole inference dataset. One thing to notice is that for training data of size $N$, the inference will be done on those $N$ prompts and is much faster than training.
> > Sft-sample-rank can also use 64 response candidates, even with tournament ranking. In Table 2, we show that rso-sample-rank is more effective than the sft-sample-rank setting with all candidates. This further demonstrates the effectiveness of our approach even with the fixed computation cost.
> >
> > **References**
> >
> > [1] https://docs.mystic.ai/docs/mistral-ai-7b-vllm-fast-inference-guide
> >
> > [2] https://github.com/ggerganov/llama.cpp/issues/3479
> >
> > [3] Pope, Reiner, et al. "Efficiently scaling transformer inference." Proceedings of Machine Learning and Systems 5 (2023).
> >
> > [4] https://github.com/ggerganov/llama.cpp/tree/master/examples/batched

---

> > ### Author Response · Authors · 2023-11-21
> >
> > >Re2: What is the specific training cost (GPU hours) for generating new samples plus the 'best of 64' to estimate M, and how long does it take to train DPO? I am aware that DPO trains quickly, so I am concerned about the potential excess time introduced by additional training.
> >
> > We first sample 64 candidates, followed by 63 pairwise reward model inference. Then we sub-select 8 responses with 4 additional pairwise reward model calls. With the same usage of TPU-v4, the relative TPU hours for rso-sample-rank (before training) is less than **10%** of that for training DPO. If we use more accelerators, the speedup on our rso-sample-rank is linear to the number of accelerators because the parallelism is across the whole training set with distributed frameworks such as Apache Beam. On the contrary, the training cannot speed up linearly because the parallelism can only happen within a batch.
> > With extra steps in rso-sample-rank, we observe strong performance gain over DPO:
> >
> > * Reddit TL;DR: **+8%** GoldRM, **+6%** AutoSxS, **2.3x** more preferred by human raters.
> >
> > * Anthropic Helpful: **+64%** GoldRM, **+71%** AutoSxS, **2.1x** more preferred by human raters.
> >
> > Based on the above evidence, we highly recommend to use RSO instead of DPO even with the extra computations.

---

> > > ### Comment · Reviewer_DcKN · 2023-11-23
> > >
> > > *'If we use more accelerators, the speedup on our rso-sample-rank is linear to'*
> > >
> > > When calculating the additional time introduced, you need to use the same amount of resources as when training dpo.
> > >
> > > In the table you provided, the Input length is 1024, while the Decode length is only 128. However, in practical chat tasks when fine-tuning LLMs, such as llama, most of the time the Decode length equals the Input length, which is 1024. Therefore, the relative training cost for rso-sample-rank (before training) is a tremendous burden for training DPO.
> > >
> > > My main concern about this article is that the idea is useful but straightforward, more like a trick needed when using dpo. In terms of improving dpo, there are many similar and more solid works in the same period.
> > >
> > > [1] Wu, Tianhao, et al. "Pairwise proximal policy optimization: Harnessing relative feedback for llm alignment." *arXiv preprint arXiv:2310.00212* (2023).
> > >
> > > [2] Li, Ziniu, et al. "ReMax: A Simple, Effective, and Efficient Method for Aligning Large Language Models." *arXiv preprint arXiv:2310.10505* (2023).
> > >
> > >
> > >
> > > My initial score is 6, and I will not change my score. But a score of 6 does not mean that I am inclined to accept; my inclination is between acceptance and rejection.

---

> ### Author Response · Authors · 2023-11-23
>
> Thanks again for further follow-up on this. Let us make a few more clarifications.
>
> >Re1: When calculating the additional time introduced, you need to use the same amount of resources as when training dpo.
>
> As stated in "With the same usage of TPU-v4, the relative TPU hours for rso-sample-rank (before training) is less than 10% of that for training DPO", we indeed use the same amount of resources. This is because training DPO is not fully parallelizable across the whole dataset. The training time increase linearly with the number of steps. For Reddit TL;DR, it takes us 22 hours to train 160k steps. For AnthropicHH, it takes us 56 hours to train 320k steps. On the contrary, all decoding are conducted on the **same** training data. Decoding is fully parallelizable across the whole dataset, because there is not parameter updates or dependencies across samples. The speed is much faster. Decoding on Reddit TL;DR and AnthropicHH takes us less than 2 hours.
>
> >Re2: In the table you provided, the Input length is 1024, while the Decode length is only 128. However, in practical chat tasks when fine-tuning LLMs, such as llama, most of the time the Decode length equals the Input length, which is 1024. Therefore, the relative training cost for rso-sample-rank (before training) is a tremendous burden for training DPO.
>
> As stated above, decoding is fully parallelizable and efficient compared with training. Even with 1024 decode length, the relative cost for rso-sample-rank before training is still much faster than the training. Batch decoding is scalable and efficient due to the nature of parallelism [1], the long decode sequence length can be further optimized (Figure 8 and Table 1 in [1]). Thus, we do not agree that rso-sample-rank before training to be a tremendous burden.
>
> > Re3: My main concern about this article is that the idea is useful but straightforward, more like a trick needed when using dpo. In terms of improving dpo, there are many similar and more solid works in the same period.
>
> I think there is a misunderstanding of the major contribution of this paper. Our major contribution is that we **identify a key problem existing in DPO** with the **response DISTRIBUTION SHIFT** between the reward model training data and the real policy to optimize. DPO adjusts the relative probability mass at two responses. If the likelihoods of both responses are low, this adjustment is not meaningful. If you have experience on running DPO, you will observer a phenomenon that it converges and overfits fast, sometimes even less than 1 epoch! The issue with that is: DPO loss can go to zero if we push down the likelihood of the negative response or the SFT likelihood is very small for the positive response (Equation (8)). rso-sample-rank solves the issue by making sure that the likelihood of the negative response is not low (because of rejection sampling) and the SFT likelihood of positive is not low (because we sample from SFT policy). Meanwhile, we unify two frameworks (SLiC and DPO) together. To address the distribution shift, we proposed the rejection sampling as a simple yet effective approach. We believe our work set a direction that future research can benefit on addressing the distribution shift problem in DPO.
>
> Regarding other research works [2][3], they are contemporaneous and we can add them in the related work in our camera-ready version.
>
> Hope we have made everything clear and we are more than happy to follow up with any further questions/concerns you may have.
>
> **References**
>
> [1] Pope, Reiner, et al. "Efficiently scaling transformer inference." Proceedings of Machine Learning and Systems 5 (2023).
>
> [2] Wu, Tianhao, et al. "Pairwise proximal policy optimization: Harnessing relative feedback for llm alignment." arXiv preprint arXiv:2310.00212 (2023).
>
> [3] Li, Ziniu, et al. "ReMax: A Simple, Effective, and Efficient Method for Aligning Large Language Models." arXiv preprint arXiv:2310.10505 (2023).

---

> > ### Comment · Reviewer_DcKN · 2023-11-23
> >
> > Why did you choose a T5-large (770M) SFT policy and a T5-XXL (11B) pairwise reward ranking model? Does this suggest that the parameters of the reward model must be significantly larger than those of the SFT model?
> >
> > *"If you have experience with running DPO, you will observe a phenomenon where it converges and overfits quickly, sometimes in less than one epoch! "*
> >
> > In my experiments with llama2 on the AnthrophicHH dataset using a cosine learning rate with 3 epoches, the accuracy was just 64%. The difference between chosen and rejected rewards kept increasing during training. With 64% traning accuracy, I don’t believe this is overfitting; it seems more like underfitting. The Reddit TL;DR task might be too simple, leading to overfitting. Therefore, I'm curious to know what the final reward accuracy of DPO and RSO was in your AnthrophicHH experiments. Similar results can be seen in the rewards_train/accuracies and rewards_train/margins on the official wandb of DPO (https://wandb.ai/eric_anthony_mitchell/dpo-demos).

---

### Official Review · Reviewer_hBgS · 2023-11-04

**Soundness:** 2 fair
**Presentation:** 3 good
**Contribution:** 2 fair
**Rating:** 5
**Confidence:** 4

**Summary:**

This paper proposes a new method to improving language model alignment using Statistical Rejection Sampling Optimization (RSO). Their method allows for more accurate estimation of the optimal policy, leading to better alignment with human preferences. Empirically, the authors demonstrate the effectiveness of RSO on two tasks: Reddit TL;DR summarization and AnthropicHH dataset. They further show that RSO works well on cross-task generalization from Reddit TL;DR. Lastly, the authors evaluate RSO using three different approaches: Reward Model, AutoSxS, and Human Evaluation. The results show that RSO variants outperform DPO and SLiC variants on both tasks.

**Strengths:**

Strengths:

- This paper introduces a new approach for improving language models alignment by exploiting rejection sampling to allow for a more precise estimation of the optimal policy. To the best of my knowledge, this proposed method is novel and well-founded. Moreover, it also include exsisting methods (e.g., DPO) as special cases.

- The paper is mostly well-written and the illustration figures are helpful for the undertsanding of this work.

- The empirical evaluation is thorough and demonstrate the effectiveness of the proposed method. Most of the claims / arguments are well-supported.

**Weaknesses:**

Weakness:

- There are still rooms for improving the presentation. For example, it would be better if the authors can explain a bit more on $\rho_{\psi}(x,y,y_b)$ when first introducing it. The authors could provide some examples or details about what would be an instantiation of $\rho_{\psi}(x,y,y_b)$.

- Step 3 in section 3.2 might be overly time-consuming and computationally expensive. Especially the set $M = \min\\{m | m\pi_{sft}(y|x) \geq \pi_{r_{\psi}}(y|x) \text{ for all } y \notin \mathcal{Y}\\}$ is expensive to compute. Rejection sampling is also slow.

- The results in Figure 3 (b) is not statistically signifcant and the confidence intervals are overlapping.

**Questions:**

1. The paper focuses on improving language models by aligning them with human preferences. I wonder how might this approach be adapted to address issues of bias and fairness in language models? Could the authors provide some additional discussions on this?

2. How does the proposed method compare against other methods, such as DPO, in terms computational efficiency?

**Details Of Ethics Concerns:**

see weakness.

---

> ### Author Response · Authors · 2023-11-13
>
> > Re1: There are still rooms for improving the presentation. For example, it would be better if the authors can explain a bit more on $\rho_{\psi}(x,y,y_b)$ when first introducing it. The authors could provide some examples or details about what would be an instantiation of $\rho_{\psi}(x,y,y_b)$.
>
> Answer: Thanks for highlighting the need for additional clarity. We admit that the presentation is not very clear at this part. In response, we have expanded the explanations of $\rho_{\psi}(x,y,y_b)$ in our revised draft. Furthermore, to illustrate its practical application, we have included specific examples of inputs and outputs, ensuring a clearer understanding of its implementation.

---

> ### Author Response · Authors · 2023-11-13
>
> > Re2: Step 3 in section 3.2 might be overly time-consuming and computationally expensive. Especially the set $M = \min ({m | m\pi_{sft}(y|x) \geq \pi_{r_{\psi}}(y|x) \text{ for all } y \notin \mathcal{Y} })$ is expensive to compute. Rejection sampling is also slow.
>
> Answer: Thank you for your insightful observation regarding the computational aspects. The reviewer is correct that directly computing $M$ can be quite expensive and rejection sampling can be slow. We clarify the following two points regarding the issue:
>
> 1. **Computing $M$**: To address this, it's important to clarify that we do not directly compute $M$. Instead, our approach involves estimating $\frac{\pi_{r_{\psi}}(y|x)}{M\pi_{\text{sft}}(y|x)}$ using 64 sequences sampled through the SFT policy, which offers a practical solution to the computational challenge you mentioned. For detailed insight into this methodology, including both the Python implementation and the algorithmic derivation, please refer to Appendix A.1 of our paper. We added a footnote to clarify this in our updated draft.
>
> 2. **Rejection sampling efficiency**:
>     * Concerning the speed of the rejection sampling process, our method begins by sampling 64 sequences via the SFT policy, followed by rejection sampling. This initial SFT sampling step is notably efficient and highly parallelizable, particularly when prompts are shared, thereby enhancing the overall computational efficiency of the process.
>     * Regarding the statistical rejection sampling algorithm, as described in Algorithm 1, it exhibits enhanced efficiency by employing a sampling-without-replacement strategy. This is achieved by excluding the selected sequences subsequent to each sampling round. Furthermore, at the commencement of each round, the maximum reward is recalculated. This recalibration ensures that, in every round, at least one sequence is invariably chosen. Specifically, the sequence whose reward is equivalent to the maximum reward is selected with a probability of one, thereby guaranteeing the selection of at least one optimal sequence in each round. This approach not only optimizes the selection process but also maintains the algorithm's effectiveness throughout its execution.
>
> We have added a section discussing the computational cost and efficiency of our approach in Appendix A.10 of the updated draft.

---

> > ### Comment · Reviewer_hBgS · 2023-11-20
> >
> > Thank you for the authors' responses. Most of my concerns and questions have been addressed by the response. However, my concern regarding the computation of "M" still persists. As stated by the authors, M is estimated by sampling "64" examples through the SFT policy. I believe this approach is quite costly and may not be a practical solution for deploying large-scale language models (e.g., >= 70B), as the inference cost would be significant. Furthermore, using samples to estimate M tends to overestimate its value. This could potentially undermine the derivation of the proposed method, RSO. Additionally, it remains unclear to me how the number of sampled sequences affects performance. Would it be possible to conduct a study similar to Figure 36 in the Anthropic paper (https://arxiv.org/pdf/2204.05862.pdf)?
> >
> > Overall, while I find the idea intriguing and the experiments well-executed, my major concern, as stated above, remains unaddressed.

---

> > > ### Author Response · Authors · 2023-11-23
> > > **Addressing Key Concerns Before Discussion Closure**
> > >
> > > We greatly appreciate your insightful comments and the opportunity to clarify our approach further. We understand the importance of addressing your concerns comprehensively before the discussion period concludes.
> > >
> > > * **Inference Cost**: We would like to emphasize the efficiency of our approach in terms of TPU/GPU resource utilization and inference time. As demonstrated in Figure 1 of [1], our method benefits from scalable and efficient batch decoding. With the extra inference cost, we observed **strong performance gains** over DPO (Table 1 and Table 4). Compared with PPO, our approach is notably more cost-effective, as it only requires a single model during training instead of four (policy, reference, value, reward), making it particularly suitable for **large-scale language models**.
> > >
> > > * **Estimation of $M$**: Regarding the estimation of $M$, our methodology doesn’t involve its direct estimation. Instead, we utilize rejection sampling to preferentially sample high-reward responses. This strategy ensures that the sampled responses align more closely with the target optimal policy, which is a crucial aspect of our approach.
> > >
> > > * **Number of Sampled Sequences**: Our experiments with 64 sampled sequences demonstrate clear improvements over sft-sample-rank, as shown in Table 1. Moreover, compared tournament ranked responses with all sampled sequences, as shown in Table 2, RSO further shows the advantage.
> > >
> > > * **Figure 36 in the Anthropic Paper [2]**: We are grateful for your suggestion to include Figure 36 from [2]. Although we are unable to incorporate this plot before the end of the discussion period, we commit to adding it in our camera-ready version. We believe this addition will provide valuable insights and further strengthen our paper.
> > >
> > > We hope that these clarifications address your concerns effectively. We are fully committed to ensuring that our research is both transparent and robust. Please do not hesitate to reach out if there are any further points you would like us to elaborate on. Your feedback is invaluable in refining our work.
> > >
> > > Thank you for your time and consideration.
> > >
> > > **References**
> > >
> > > [1] Pope, Reiner, et al. "Efficiently scaling transformer inference." Proceedings of Machine Learning and Systems 5 (2023).
> > >
> > > [2] Bai, Yuntao, et al. "Training a helpful and harmless assistant with reinforcement learning from human feedback." arXiv preprint arXiv:2204.05862 (2022).

---

> ### Author Response · Authors · 2023-11-13
>
> > Re3: The results in Figure 3 (b) is not statistically signifcant and the confidence intervals are overlapping.
>
> Answer: Thank you for your insightful feedback regarding the statistical significance and confidence intervals in Figure 3(b). We appreciate the opportunity to clarify and reinforce the findings of our study.
>
> 1. **Purpose of the Plot**: The primary objective of Figure 3(b) is to illustrate our process of hyper-parameter selection, focusing exclusively on the proxy reward model. This model's win rate against the SFT targets is depicted on the y-axis. It's crucial to note that the plot is to guide hyper-parameter choices.
>
> 2. **Comparison among loss functions**: We don’t claim the preferred choice between sigmoid-norm and hinge-norm loss functions. Regarding hinge loss, it performs the best (not statistically significantly) because hinge loss ignores the SFT policy completely and fully trusts the preference order determined by the proxy reward model. That explains why the proxy reward model win rate is high.
>
> 3. **Significance of rso-sample-rank**: Despite the overlapping confidence intervals, a critical observation from the plot is that the rso-sample-rank consistently outperforms sft-sample-rank across all three loss functions. This is visually represented by the dotted lines (sft-sample-rank) remaining outside the shaded regions of all rso-sample-rank curves.
>
> 4. **Evaluating top-k-over-N and Beta Selection**: The plot also challenges the notion that top-k-over-N is universally optimal. When beta equals zero, rso-sample-rank aligns with top-k-over-N. Interestingly, we observe that a beta of 0.5 achieves a higher proxy reward model win rate than beta=0. Although this difference is not statistically significant, it indicates that a nuanced approach to beta selection could yield better results than the default top-k-over-N strategy.
>
> 5. **Inclusion of Additional Baselines**: To robustly validate our findings and address the concern of statistical significance, we have incorporated additional baselines, namely RAFT [1] and ReST [2], into our revised draft. These inclusions will provide a broader comparative framework, enhancing the validity and reliability of our conclusions.
>
> **References**
>
> [1] Dong, Hanze, et al. "Raft: Reward ranked finetuning for generative foundation model alignment." arXiv preprint arXiv:2304.06767 (2023).
>
> [2] Gulcehre, Caglar, et al. "Reinforced self-training (rest) for language modeling." arXiv preprint arXiv:2308.08998 (2023).

---

> ### Author Response · Authors · 2023-11-13
>
> > Re4: The paper focuses on improving language models by aligning them with human preferences. I wonder how might this approach be adapted to address issues of bias and fairness in language models? Could the authors provide some additional discussions on this?
>
> Answer: Thank you for highlighting the critical issues of bias and fairness in language models. We fully appreciate their significance in the broader context of AI ethics.
>
> (1) “I wonder how might this approach be adapted to address issues of bias and fairness in language models?”
>
> Our current research focuses on preference alignment in language models. In practical scenarios, reward scores are often multi-dimensional, and the aim of alignment is to attain a Pareto optimal frontier [1]. This allows for the introduction of additional objectives such as harmlessness, safety, and bias preference pairs. Our method is adaptable, functioning with either weighted-averaged reward scores or through integration with multi-objective DPO loss functions [2]. Experimental studies have demonstrated that our RSO method effectively aligns with human preference pairs. We posit that our approach has the potential to enhance fairness and reduce bias in language models, provided it is applied with appropriate human preference pairs. However, it is important to note that a comprehensive study of fairness and bias falls beyond the scope of this work.
>
> (2) “Could the authors provide some additional discussions on this?”
>
> We have added Appendix A.9 for a deeper examination of bias and fairness in language models.
>
>
> **References**
>
> [1] Bai, Yuntao, et al. "Constitutional ai: Harmlessness from ai feedback." arXiv preprint arXiv:2212.08073 (2022).
>
> [2] Zhou, Zhanhui, et al. "Beyond One-Preference-for-All: Multi-Objective Direct Preference Optimization." arXiv preprint arXiv:2310.03708 (2023).

---

> ### Author Response · Authors · 2023-11-13
>
> > Re5: How does the proposed method compare against other methods, such as DPO, in terms computational efficiency?
>
> Answer: Thank you for raising the computational complexity question. In terms of computational efficiency, our approach is closer to SLiC-HF-sample-rank than DPO. We list the comparisons with other algorithms as follows:
>
> * Compared to DPO and SLiC-HF-direct, our approach requires an additional trained pairwise reward model, and also sample and rank stages. The pairwise reward model training is a standard text-to-text model without Bradley-Terry model assumption. The sample and rank are scalable and efficient with increased number of servers, especially with shared prompts in SFT sample and short decoding length with reward model.
>
> * Compared to SLiC-HF-sample-rank, we need to sample more sequences from SFT policy as candidates, followed by a rejection sampling algorithm. Both steps are scalable and efficient as described in response to reviewer hBgS.
>
> * Compared to RLHF, RSO is offline with efficient memory usage (only one copy of policy network instead of four copies in PPO: policy, reference, value, reward), and the sample-rank stage is fully parallelizable (no need to conduct the online sample, which is not parallelizable).
>
> To clarify this, we added Appendix A.10 for thorough discussion in our updated draft.

---

> ### Author Response · Authors · 2023-11-20
>
> > Re1: However, my concern regarding the computation of "M" still persists. As stated by the authors, M is estimated by sampling "64" examples through the SFT policy. I believe this approach is quite costly and may not be a practical solution for deploying large-scale language models (e.g., >= 70B), as the inference cost would be significant.
>
> Answer: thank you for your follow-up questions. Sampling decoded sequences on large-scale language models seems to be costly.  However, 64 is not a big value compared to some existing best-of-n studies (Fig.8 in LLAMA2 [1], Sec 3.1 in [2]). Even for large-scale language models, 64 samples used in our experiment are not expensive and very achievable in practical settings based on a few reasons:
>
> 1. **Parallelism of the algorithm**: unlike the PPO algorithm that can only be parallelized within each batch, RSO can be parallelized across the whole training set. Accelerators like GPU, TPU are very good at parallel computations and are scalable linearly with more servers for inference. Public libraries like llama.cpp [3] support this feature.
>
> 2. **Prompt caching**: prompt caching is supported in modern LLM servers [4][5]. To sample 64 responses, the input query/key/value embeddings can be pre-computed in a parallelized way and cached in memory. The sampling cost will be smaller to generate 64 responses from one prompt versus generating them from different prompts. Moreover, batching prompts can further be sped up as shown in [5] and total memory/time turns out to be (significantly) sublinear as one would expect (Figure 1 in [5]).
>
> Compared with online algorithms such as PPO, we save three networks (reference, value, reward) during policy training, which saves up to 3x memory usage. Furthermore, RSO is fully parallelizable on decoding and reward inference on the whole training data.
>
> **References**
>
> [1] Touvron, Hugo, et al. "Llama 2: Open foundation and fine-tuned chat models." arXiv preprint arXiv:2307.09288 (2023).
>
> [2] Gao, Leo, John Schulman, and Jacob Hilton. "Scaling laws for reward model overoptimization." International Conference on Machine Learning. PMLR, 2023.
>
> [3] https://github.com/ggerganov/llama.cpp/tree/master/examples/batched
>
> [4] Pope, Reiner, et al. "Efficiently scaling transformer inference." Proceedings of Machine Learning and Systems 5 (2023).
>
> [5] Kwon, Woosuk, et al. "Efficient memory management for large language model serving with pagedattention." Proceedings of the 29th Symposium on Operating Systems Principles. 2023.

---

> ### Author Response · Authors · 2023-11-20
>
> >Re2: Furthermore, using samples to estimate M tends to overestimate its value. This could potentially undermine the derivation of the proposed method, RSO. Additionally, it remains unclear to me how the number of sampled sequences affects performance. Would it be possible to conduct a study similar to Figure 36 in the Anthropic paper (https://arxiv.org/pdf/2204.05862.pdf)?
>
> Answer: thanks again for raising the above important points.
>
> (1) "Furthermore, using samples to estimate M tends to overestimate its value. This could potentially undermine the derivation of the proposed method, RSO."
>
> We agree that the estimation of $M$ is biased given limited response candidates. We show that in Theorem 1, as $M$ goes to infinity, we can sample the responses from the exact optimal policy. In practice, $M$ is finite and our algorithm is indeed an approximation. There are two evidence we would like to emphasize regarding this:
>
> 1. **The rejection sampling algorithm**: from the Algorithm 1 we can see, at each iteration in the while loop, the highest remaining response will always be selected (unless $\beta=\infty$, which goes back to sft-sample-rank scenario). So we ensure that the responses always contain high reward ones. From that point of view, we include the traditional rejection sampling as a special case and our bias goes towards that. On the other hand, as $\beta=\infty$, Algorithm 1 will sub-sample random responses from the SFT generated candidates. So on two extremes, our algorithm is well calibrated and behaved. For other $\beta$'s, the sampled responses will follow a distribution that is closer to the optimal policy than the SFT one. We can intuitively observe that from Figure 2 in the paper.
>
> 2. **Practical evidence**: from Figure 3(b) and Table 1-4, we can conclude that rso-sample-rank shows consistent gains over sft-sample-rank, which suggests that the rejection sampled responses are indeed from a better distribution than SFT.
>
>
> (2) “Additionally, it remains unclear to me how the number of sampled sequences affects performance. Would it be possible to conduct a study similar to Figure 36 in the Anthropic paper (https://arxiv.org/pdf/2204.05862.pdf)?”
>
> This is indeed a great suggestion. We may not have enough time to conduct this study given the limited time remaining before rebuttal ends. But we can seriously consider adding it in the camera-ready version.

---

### Author Response · Authors · 2023-11-13

We express our sincere gratitude for your comprehensive and insightful feedback. Your comments have been instrumental in refining our work. We are pleased that the novelty and comprehensiveness and rigor of our experiments have been recognized. Following your suggestions, we have conducted additional experiments and made significant enhancements to our manuscript, which we believe address your concerns effectively.

1. Inclusion of New Baselines:
    * Reviewer nTnq's Suggestion: We have integrated two additional baselines for rejection sampling - RAFT [1] and ReST [2], as recommended.
    * Outcome: As shown in the following table, our method (RSO) demonstrates superior performance over these robust baselines, underscoring RSO's effectiveness.

| Task          | Approach           | Proxy Reward (%) | Gold Reward (%) | AutoSxS (%) |
| ------------- | ------------------ | ---------------- | --------------- | ----------- |
| Reddit TL; DR | RAFT               | 74.84            | 68.51           | 53.77       |
| | ReST               | 49.03            | 46.17           | 34.36       |
|   | $RSO_{sigmoid-norm}$ | 92.37            | 82.22           | **71.86**       |
|   | $RSO_{hinge-norm}$   | **92.80**            | **83.45**           | 70.84       |

| Task          | Approach           | Proxy Reward (%) | Gold Reward (%) | AutoSxS (%) |
| ------------- | ------------------ | ---------------- | --------------- | ----------- |
| AnthropicHH   | RAFT               | 58.21            | 40.00           | 24.99       |
|                        | ReST               | 43.48            | 30.33           | 15.58       |
|                        | $RSO_{sigmoid-norm}$ | **86.94**            | **59.15**  | **40.98** |
|                       | $RSO_{hinge-norm}$  | 84.44            | 57.75 | 38.58       |

2. Gold Reward Model as Evaluation Metrics:
    * Reviewer DcKN's Suggestion: In line with your guidance, we trained PaLM-2-S pairwise reward models using the protocol from [3].
    * Outcome: This new gold reward evaluation method not only complements our existing methodologies but also reaffirms RSO's superiority across all studied approaches. The win rates and detailed analyses are included in our revised manuscript.

3. Additional Appendices for Comprehensive Understanding:
    * Appendix on Bias and Fairness: Addressing Reviewer hBgS's suggestion, Appendix A.9 delves into the critical issues of bias and fairness in language models, enhancing the depth of our discussion on these pivotal aspects.
    * Appendix on Computational Efficiency: In response to Reviewer hBgS, Appendix A.10 is dedicated to a thorough comparison of computational efficiency, providing a clearer understanding of the practical implications of our approach.

These improvements, we believe, significantly enhance the quality and impact of our work. We are confident that the revisions made not only address your concerns but also contribute meaningfully to the field.
We appreciate your consideration and look forward to any further suggestions that could improve our work.

**References**

[1] Dong, Hanze, et al. "Raft: Reward ranked finetuning for generative foundation model alignment." arXiv preprint arXiv:2304.06767 (2023).

[2] Gulcehre, Caglar, et al. "Reinforced self-training (rest) for language modeling." arXiv preprint arXiv:2308.08998 (2023).

[3] Gao, Leo, John Schulman, and Jacob Hilton. "Scaling laws for reward model overoptimization." International Conference on Machine Learning. PMLR, 2023.

---

### Public Comment · ~Wei_Xiong9 · 2023-12-08
**nice work on clarification for a misunderstanding in the literature**

I noticed this paper when I go through the works related to alignment in the submissions of ICLR, and it is also my pleasure to see that one of my works has been used as the standard baseline for algorithm comparison. In general, I enjoyed reading this nice work.

I have been confused by the derivation of DPO from equation (6) -> (7) in the original paper for quite some time and this paper clearly states this tricky gap from a statistical viewpoint. Personally, I believe that pointing out such a technical gap in a popular algorithm with large impact in the community is interesting and very relevant to the topic of ICLR. Besides, I noticed that there has been considerable attentions on the debate of whether we need reward modeling, as well as RL (PPO), in view of the success of DPO. I believe that this paper does support the standard RLHF framework outlined in llama2 report and instruct-gpt paper, because the reward modeling is helpful to mitigate the distribution shift issue between our target distribution and the data distribution if we adopt the algorithm proposed in this paper (i.e. RSO).

---

### Meta-Review · Area_Chair_R2jJ · 2023-12-10

**Metareview:**

The reviewers were split about this paper and did not come to a consensus: on one hand they appreciated the principled approach and the thorough empirical investigation; on the other they had doubts about (1) the computational expense, and (2) the optimality of the approach. After going through the paper and the discussion I have decided to vote to accept because the authors respond convincingly to each of the main concerns. Specifically for (1), the reviewers argued that 64 sampled sequences for rejection sampling could be computationally prohibitive for LLMs. The authors responded that these can be parallelized for shared prompts and that their sampling scheme without replacement combined with reward recalculation ensures that one sequence is always chosen. Further they point out that, compared to similar work ReST, for a specific hyperparameter setting RSO will need roughly the same number of samples as ReST. Compared to k-grow-round ReST, RSO is k times more computationally efficient. I believe this point is resolved, however, I would strongly urge the authors to run an experimental comparison that plots a metric (e.g., Gold Reward) vs. time, and run each method for different sized budgets, to see how the curves of performance vs. time look for each method. For (2), the reviewers argued that the policy induced by RSO is not optimal, contrary to the claim made in the paper. The authors agree that this is true and renamed ‘optimal policy’ to ‘estimated optimal policy’. They also argue that their estimation has advantages over DPO, i.e., not requiring a rank-1 approximation to preference information, and utilizes a text-to-text classification model instead of a generation task. This resolves the reviewers concern and does not detract from the method proposed in the paper, but clarifies the technical aspects more clearly. Overall, the authors have convinced me that they are able to fix these concerns in the final version in their responses to reviewers. For these reasons I argue for acceptance. Authors: the reviewers have given extremely detailed feedback and I recommend the authors follow / respond to their comments closely, as you have already started to do. Once this is done, the paper will make a great contribution to the conference!

**Justification For Why Not Higher Score:**

The paper needed serious editing in response to reviewer concerns, for this reason I do not think it should be given a higher score.

**Justification For Why Not Lower Score:**

The authors did a very detailed job during the discussion period responding to reviewers (if a bit too verbose). It's unclear to me why reviewers did not increase their scores further.

---

### Decision · Program_Chairs · 2024-01-16

Accept (poster)